# Unlocking Phytate with Phytase: A Meta-Analytic View of Meat-Type Chicken Muscle Growth and Bone Mineralization Potential

**DOI:** 10.3390/ani14142090

**Published:** 2024-07-17

**Authors:** Emmanuel Nuamah, Utibe Mfon Okon, Eungyeong Jeong, Yejin Mun, Inhyeok Cheon, Byungho Chae, Frederick Nii Ako Odoi, Dong-wook Kim, Nag-Jin Choi

**Affiliations:** 1Department of Animal Science, Jeonbuk National University, Jeonju 54896, Republic of Korea; kingjs664@jbnu.ac.kr (E.J.); 202012536@jbnu.ac.kr (Y.M.); cih6770@jbnu.ac.kr (I.C.); byungho721@jbnu.ac.kr (B.C.); 2Department of Animal Science, Faculty of Agriculture, Akwa Ibom State University, Mkpat Enin 532111, Nigeria; utibeokon93@gmail.com; 3Department of Animal Science, School of Agriculture, University of Cape Coast, Cape Coast CC 3321, Ghana; fodoi@ucc.edu.gh; 4Department of Animal Science, Korea National University of Agriculture and Fisheries, Jeonju 54874, Republic of Korea; poultry98@korea.kr

**Keywords:** broilers, calcium, exogenous enzyme, growth performance, phosphorus, mineral utilization, welfare

## Abstract

**Simple Summary:**

Phytase has been utilized extensively to free dietary phytate P and reduce environmental pollution. In commercial broiler production, phytase has been demonstrated to provide several benefits, including improved growth performance and the bioavailability of nutrients while enhancing their utilization. Yet, the multifaceted factors facilitating phytase’s pronounced effects still need to be fully explored. Therefore, this study sought to summarize the impact of supplemental phytase in P- and Ca-deficient diets on broiler growth performance, bone strength, and mineralization, and explore the factors contributing to the variation across studies. The meta-analysis revealed that phytase supplementation in P- and Ca-deficient diets improves average daily feed intake, weight gain, feed conversion ratio, tibia ash, P and Ca, and bone strength in broilers across the growth phases. However, the effects were more pronounced in starter birds. In terms of improvement, the magnitude of the outcomes in P- and Ca-deficient diet supplementation with phytase was substantially associated with age, broiler strain, dietary P source, and duration of supplementation. In conclusion, phytase enzyme supplementation over a long period ameliorates the adverse effects of P- and Ca-deficient diets to improve growth performance, mineralization, and strength of the bones.

**Abstract:**

The inclusion of exogenous phytase in P- and Ca-deficient diets of broilers to address the growing concern about excessive P excretion into the environment over the years has been remarkably documented. However, responses among these studies have been inconsistent because of the several factors affecting P utilization. For this reason, a systematic review with a meta-analysis of results from forty-one studies published from 2000 to February 2024 was evaluated to achieve the following: (1) quantitatively summarize the size of phytase effect on growth performance, bone strength and mineralization in broilers fed diets deficient in P and Ca and (2) estimate and explore the heterogeneity in the effect size of outcomes using subgroup and meta-regression analyses. The quality of the included studies was assessed using the Cochrane Collaboration’s SYRCLE risk of bias checklists for animal studies. Applying the random effects models, Hedges’ g effect size of supplemented phytase was calculated using the R software (version 4.3.3, Angel Food Cake) to determine the standardized mean difference (SMD) at a 95% confidence interval. Subgroup analysis and meta-regression were used to further explore the effect size heterogeneity (*P*_SMD_ ≤ 0.05, *I*^2^ > 50%, n ≥ 10). The meta-analysis showed that supplemental phytase increases ADFI and BWG and improves FCR at each time point of growth (*p* < 0.0001). Additionally, phytase supplementation consistently increased tibia ash, P and Ca, and bone strength (*p* < 0.0001) of broilers fed P- and Ca-deficient diets. The results of the subgroup and meta-regression analyses showed that the age and strain of broiler, dietary P source, and the duration of phytase exposure significantly influence the effect size of phytase on growth and bone parameters. In conclusion, phytase can attenuate the effect of reducing dietary-available phosphorus and calcium and improve ADFI, BWG, and FCR, especially when added to starter diets. It further enhances bone ash, bone mineralization, and the bone-breaking strength of broilers, even though the effects of bone ash and strength can be maximized in the starter phase of growth. However, the effect sizes of phytase were related to the age and strain of the broiler, dietary P source, and the duration of phytase exposure rather than the dosage.

## 1. Introduction

Calcium and phosphorus, constituting the first (99%) and second (80%) stored minerals in the skeleton as hydroxyapatite, are vital for numerous essential functions [1]. Their presence in balanced proportions within diets is equally crucial. These macro minerals, whether fed separately or in conjunction, are pivotal in facilitating bone formation and fulfilling various metabolic requirements [2]. Yet, phosphorus (P) remains one of the widely researched nutrients in monogastric diets due to its importance in several biochemical reactions and its debilitating environmental effects if not managed properly [3]. Non-ruminant species, especially poultry and pigs, are known to be efficient feed converters to muscle but cannot utilize phosphorus (P) from plant-based feed ingredients sufficiently [4,5,6], with significant amounts of P excreted in their waste [7]. The peculiarity is that most of these feed ingredients constituting a substantial portion of the broiler diet are from the cereals and oilseed families [8], with P in the form of phytin P [9]. This P, approximately 50–85%, is bound as the salt form of phytic acid, called phytate [10,11,12]. 

Phytate, a phosphorus storage form in plants, not only serves as both a nutrient and an antioxidant but also acts as an anti-nutrient. It is known to be less readily available to non-ruminants, like broilers [5,6]. Within the small intestines, phytate forms robust chelates with cations such as Ca^2+^ or Cu^2+^, creating insoluble complexes that impede the effectiveness of endogenous phytase in the avian gut. This, in turn, decreases their capacity to release P [3,10,11,12] and promotes interference with protein digestion [13]. Due to this reason, inorganic phosphate became necessary to tackle the low P bioavailability issue and satisfy the P requirement of poultry [14]. Adding inorganic P, an expensive and finite mineral resource, to diets that may already contain sufficient P, though in an unavailable form, results in total P levels above the requirement [14]. Consequently, much of the excess P in the diet is excreted in animal waste [15]. As excreted P is a function of total P, heightened concerns in the past decade over the phosphorus pollution of the environment presented policies about P regulation in animal feeding operations [14,16,17]. Following the increasing awareness and concerns, cut downs on “insurance” P levels in animal feeds prompted poultry nutritionists to supplement feed additives, such as phytase, in the nutrition of non-ruminants [18,19,20,21].

Phytase efficacy to liberate phytate-bound phosphorus and improve the bioavailability of phosphorus in poultry feed is well-known [22,23]. Supplementation with commercial exogenous phytase, a common practice in poultry nutrition, is established to reduce the addition of inorganic phosphorus (P) and calcium (Ca) in poultry feed, improve growth performance, nutrient utilization, bone mineralization, and increase amino acid digestibility [24,25] at diverse physiological stages, releasing P and another mineral (e.g., Ca, Zn, Fe, Cu) [8], as well as reducing environmental P pollution [4]. However, wide variations in supplemented phytase effectiveness on P availability, even at the same phytase dose and diet type, are reported to contribute to unreliable diet formulation [26]. More recent studies have indicated that the release and utilization of P due to supplemented phytase is variable, depending on the species, age, and physiological conditions of the birds, as well as dietary factors like the source of phytate, phytate concentration, mineral concentrations, and the dosing and feeding duration of phytase [27,28,29,30,31,32]. 

The effects of supplemental phytase in diets low in P and Ca on growth performance, bone strength, and mineralization in broiler chickens have been extensively studied [33,34,35,36,37,38,39,40]. However, these studies were conducted under different conditions, making generalizations from such results difficult. Meta-analysis (MA), a statistical method, combines numerical summary results of independent studies/trials to pool an estimate of the effect size measures [41]. Compared to that reported in individual studies, when the results of independent studies are conflicting, MA arrives at a better estimate of the population effect size. Its use in exogenous phytase (EP) studies in broilers has focused on non-phytate phosphorus, phosphorus retention, interactive effects of Ca, vitamin D_3,_ and *Eimeria* infections [42,43,44,45,46]. As the genetic framework of broilers, new phytases, and research continue to proliferate in this domain, evaluating P utilization in current broiler chickens’ strains, dietary P source, and physiological stage is imperative. Therefore, the objectives of this study were to (1) summarize the size of exogenous phytase effect on growth performance, bone strength, and mineralization in broilers fed P- and Ca-deficient basal diets and (2) estimate and explore the heterogeneity of the effect sizes of outcomes using subgroup and meta-regression analyses. Hence, we hypothesized that reducing P and Ca in a diet supplemented with phytase would not compromise broilers’ performance, bone strength, and mineralization at different growth phases.

## 2. Materials and Methods

### 2.1. Literature Search and Study Screening

To thoroughly frame and construct the right testable questions, the Population, Interventions, Comparators, and Outcomes (PICO) elements suggested by Schmid et al. [47] motivated the foundation of the review topic, the protocol, and the strategy for searching the relevant literature. In identifying the PICO elements, Population (broiler chicken), Interventions (varying doses of exogenous phytase supplemented to basal standalone P- and Ca-deficient diet), Comparators (standalone P- and Ca-deficient basal diet), and Outcomes (response variables including growth performance, bone strength, and mineralization) were established. Figure 1 illustrates how the meta-analysis, registered under the number INPLASY20240096, identified, selected, and included information in accordance with the most recent Preferred Reporting Items for Systematic Reviews and Meta-Analysis (PRISMA) standards [48]. 

To address the research question “exogenous phytase phytate-P unlocking efficacy for potential muscle growth and bone mineralization in broiler at different growth phases”, a comprehensive literature search for articles published between 2000 and February 2024 was conducted using Web of Science (retrieved on 21 February 2024), Scopus (retrieved on 22 February 2024), ScienceDirect (retrieved on 21 February 2024), PubMed (retrieved on 22 February 2024), and Google Scholar (retrieved on 20 February 2024) online databases. Additionally, we searched Poultry Science (retrieved on 21 February 2024), a leading journal in poultry research, for relevant articles. In all the databases and Poultry Science journal, the keywords “phytase supplementation”, “phosphorus”, “broiler chicken”, “growth”, “bone mineralization”, and “blood characteristics” were used.

### 2.2. Criteria for Inclusion and Exclusion

Upon pooling the search results from all five databases and the Journal of Poultry Science (1305) in Zotero (Version 6.0.36), duplicate articles were eliminated. To ensure the appropriateness of the remaining articles in addressing the current research question, two reviewers independently and thoroughly applied a two-step screening process. First, screening was performed using the title and abstract to exclude review papers, conference papers, stimulated studies (in vitro), and studies that did not include broilers and did not measure the response variables of interest. In the second step, papers that made it beyond the title and abstract screening were evaluated for eligibility using the meta-analysis’s inclusion and exclusion criteria. Criteria for inclusion comprised studies (1) published in a peer-reviewed English journal, (2) that had a standalone P- and Ca-deficient basal diet supplemented with phytase, (3) that used broiler strains at either starter or grower-finisher phase, (4) that allotted broilers using randomization, (5) quantified phytase dose, (6) that displayed the means of the control and experimental group along with variability measures (standard deviation or standard error of mean) and sample size, and (7) registered the parameters of interest. Studies that (1) were challenged, (2) provided phytase as a replacement in the diet of starter and grower-finisher broilers, and (4) combined phytase with other exogenous enzymes or additives were excluded from consideration. Forty-one (41) full-text articles were used for the meta-analysis, as illustrated in Figure 1. 

### 2.3. Extraction of Data

After screening, two team members independently extracted two main sets of data from studies identified as relevant (Table 1). The characteristics and design features of studies including author, year, country, breed/strain, phase of growth, basal diet source of phosphorus and calculated chemical composition of the standalone P- and Ca-deficient basal diet, category of phytase, origin of phytase, phytase expression host, dosage of phytase, and supplementation duration were extracted from each paper. The outcomes of interest were derived from two categories. The average daily feed intake (ADFI), average daily gain (ADG), body weight gain (BWG), and feed conversion ratio (FCR) were all included in the growth performance category. The second category, bone strength and mineralization, included bone-breaking strength (BBS), tibia ash (TA), tibia calcium (TCa), and tibia phosphorous (TP). The mean and standard deviation of all outcomes were extracted from each paper for the registered phytase and control groups. In papers that included several phytase dose supplements, each treatment comparison with the control group was considered a distinct trial. The retrieved data from eligible papers were compiled and constructed into a database using structured spreadsheets created in Google Sheets (Google LLC, Mountain View, CA, USA).

### 2.4. Appraisal of Study and Risk Assessment for Bias

Two team members independently evaluated the quality, validity, and potential risk of bias of the eligible studies using the risk of bias (RoB) checklists of items for animal studies developed by the Cochrane Collaboration’s Systematic Review Center for Laboratory Animal Experimentation (SYRCLE) [49]. The appraisal of eligible papers was related to bias indicators, including selection bias (random sequence generation, baseline characteristics, and allocation concealment), performance bias (random housing and blinding of participants and personnel), detection bias (random outcome assessment), attrition bias (incomplete outcome data), reporting bias (selective reporting), and other biases. In situations where disagreements over the assessment existed, further discussions with a third member were sorted.

### 2.5. Statistical Analysis

#### 2.5.1. Meta-Analysis (MA)

Using the “meta” and “metafor” packages of R (version 4.3.3, “Angel Food Cake,” R Foundation for Statistical Computing Platform, Vienna, Austria), sixteen (16) separate meta-analyses were performed to combine estimates of phytase supplementation on growth performance, bone strength, and mineralization in broilers across studies. With procedures of the random effects models, the means of the experimental units (control and treatment) were recorded as continuous data, and their impact was computed using Hedges’ g as the standardized mean difference (SMD), commonly known as the effect size (ES). The standard deviation (SD) of the group means with and without exogenous phytase was used to normalize the difference between the means of the treated and control groups. Since the random effects model is more conservative than the fixed effects model, it was utilized to estimate the effect size [50,51]. At a confidence interval (CI) of 95%, a calculated SMD with a *p*-value ≤ 0.05 was declared statistically significant.

#### 2.5.2. Heterogeneity Assessment

The chi-square test (Q) and the *I*^2^ statistics were used to evaluate and quantify the effect size heterogeneity. With relatively low power, the Q test of a *p*-value of ≤0.10 was considered to have significant heterogeneity [52]. The *I*^2^ statistic, which expresses the proportion of between-study variability, on the other hand, was assessed using values ranging from 0 to 100%. Using the Borenstein et al. [53] benchmark range for *I*^2^ statistics, *I*^2^ < 25%, 25% ≤ *I*^2^ ≤ 50%, 50% ≤ *I*^2^ < 75%, and 75% ≤ *I*^2^ ≤ 100% were interpreted as representing low, moderate, high, and very high levels of heterogeneity, respectively. In cases where the MA revealed significant heterogeneity (Q test *p*-value of ≤0.10 and *I*^2^ > 50%) in the investigated outcomes, the sources of heterogeneity between trials were identified using meta-ANOVA and meta-regression analyses.

**Table 1 animals-14-02090-t001:** A synopsis of the attributes of the eligible papers.

Studies	Country	Broiler Strain	Dietary P Source ^1^	Origin	Phytase Expression Host	Category of Phytase
Ajuwon et al. [54]	Germany	Ross 308	Corn–Soybean	Bacteria	*Escherichia coli*	6-phytase
Beeson et al. [4]	UK	Ross 308	Corn–Soybean	Bacteria	*Escherichia coli*	6-phytase
Bello et al. [55]	Canada	Ross 308	Corn–Soybean	Bacteria	*Buttiauxella* sp. *Citrobacter braakii*	6-phytase 6-phytase
Borda-Molina et al. [56]	Germany	Ross 308	Corn–Soybean	Bacteria	*Escherichia coli*	6-phytase
Bowen et al. [57]	USA	Ross 708	Corn–Soybean	Bacteria	*Escherichia coli*	6-phytase
Broch et al. [33]	Brazil	Cobb 500	Corn–Soybean	Bacteria	*Escherichia coli*	6-phytase
Broch et al. [34]	Brazil	Cobb 500	Corn–Soybean	Fungi	*Aspergillus oryzae*	3-phytase
Broch et al. [58]	Brazil	Cobb 500	Corn–Soybean	Fungi	*--*	3-phytase
Campasino et al. [35]	USA	Hubbard × Cobb 500	Corn–Soybean	Fungi	*Aspergillus oryzae*	6-phytase
Cowieson et al. [36]	UK	Ross	Corn–Soybean	Fungi	*Aspergillus oryzae*	6-phytase
Cowieson et al. [37]	Poland	Ross 308	WCS	Bacteria	*Escherichia coli*	6-phytase
Dersjant-Li et al. [59]	USA	Cobb 500	Corn–Soybean	Bacteria	*Buttiauxella* sp.	6-phytase
Dersjant-Li et al. [38]	New Zealand	Ross 308	CSRRB	Bacteria	*Trichoderma reesei*	6-phytase
Dessimoni et al. [39]	Brazil	Cobb 500	Corn–Soybean	Bacteria	*Escherichia coli*	6-phytase
Ennis et al. [60]	USA	Ross × Ross 708	Corn–Soybean	Bacteria Bacteria Fungi	*Escherichia coli* *Buttiauxella* sp. *Aspergillus niger*	6-phytase 6-phytase 3-phytase
Gehring et al. [61]	USA	Ross × Ross 708	WCS	Fungi	*Aspergillus niger*	6-phytase
Ghahri et al. [62]	Iran	Ross 308	Corn–Soy–DDGS	Bacteria	*Escherichia coli*	6-phytase
Gulizia et al. [40]	USA	Ross 708 × YPM	Corn–Soybean	Bacteria	*Escherichia coli*	6-phytase
Hernandez et al. [63]	USA	Cobb 500	Corn–Soybean	Bacteria	*Escherichia coli*	6-phytase
Houshyar et al. [64]	Iran	Ross 308	WCS	Bacteria	*Serratia odorifera*	3-phytase
Javadi eta al. [65]	Spain	Ross	WCS	Bacteria	*Trichoderma reesei*	6-phytase
Jlali et al. [66]	France	Ross 308	Wheat–Soybean	Bacteria	*Escherichia coli*	6-phytase
Józefiak et al. [67]	Poland	Ross 308	Corn–Soybean	Bacteria	*Escherichia coli*	6-phytase
Karami et al. [68]	Germany	Ross 308	Corn–Soybean	Bacteria	*Buttiauxella* sp.	6-phytase
Kiarie et al. [69]	Canada	Ross 308	Corn–Soybean	Bacteria	*Escherichia coli*	6-phytase
Kriseldi et al. [70]	USA	Yield Plus × Ross 708	Corn–Soybean	Bacteria	*Escherichia coli*	6-phytase
Kwon et al. [71]	Korea	Ross 308	Corn–Soybean	Bacteria	*Escherichia coli*	6-phytase
Liu et al. [72]	Australia	Ross 308	Corn–Soybean	Fungi	*Aspergillus oryzae*	6-phytase
Moita et al. [73]	USA	Ross 308	Corn–Soybean	Fungi	*Aspergillus niger*	6-phytase
Powell et al. [74]	USA	Ross × Ross 508	Corn–Soybean	Bacteria	*Escherichia coli*	6-phytase
Ptak et al. [75]	Poland	Ross 308	WRES	Fungi Bacteria	*Aspergillus ficcum* *Escherichia coli*	3-phytase 6-phytase
Ptak et al. [76]	Poland	Ross 308	WRES	Bacteria	*Escherichia coli*	6-phytase
Shang et al. [77]	Canada	Ross × Ross 308	WCS	Bacteria	--	6-phytase
Shi et al. [78]	USA	Cobb 500	Corn–Soybean	Bacteria	*Buttiauxella* sp.	6-phytase
Walk et al. [79]	UK	Ross 308	Wheat–Soybean	Bacteria	*Escherichia coli*	6-phytase
Walk et al. [13]	USA	Cobb 500	Corn–Soybean	Bacteria	*Escherichia coli*	6-phytase
Walk and Olukosi, [80]	UK	Ross 308	Wheat–Soybean	Bacteria	*Escherichia coli*	6-phytase
Walk and Poernama, [81]	UK	Lohman Indian River Straight-run	Corn–Soybean	Bacteria	*Escherichia coli*	6-phytase
Walk et al. [82]	USA	Cobb 500	Corn–Soybean	Bacteria	*Escherichia coli*	6-phytase
Woyengo et al. [83]	Canada	Ross	Corn–Soybean	Bacteria	*--*	6-phytase
Zhang et al. [84]	China	Cobb 500	Corn–Soybean	Bacteria	*Escherichia coli*	6-phytase
	China	Ross 308	Corn–Soybean	Bacteria	*Citrobacter braakii*	6-phytase

^1^ WCS: wheat–corn–soybean; CSRRB: corn–soybean–rapeseed–rice bran; WRES: wheat–rapeseed expeller–soybean.

#### 2.5.3. Publication Bias

Funnel plots were utilized to illustrate the bias, while Egger’s linear test was used to precisely assess publication bias (PB) using numerical data [85]. Since PB can produce false-positive claims, tests evaluating it can only be successful when the variable to be assessed has at least ten studies, and a substantial (*p* ≤ 0.05) bias is detected. As a result, Egger’s test and funnel plots were limited to variables that satisfied the requirements. The number of potential missing observations was estimated using Duval and Tweedie’s “trim-and-fill” approach [86], in situations where the statistical evidence of PB was detected. 

#### 2.5.4. Meta-ANOVA (Subgroup Analysis) and Meta-Regression

Study-level categorical covariates, including basal diets’ dietary P source and broiler strains, were subjected to subgroup analysis. Meta-regression analysis, on the other hand, was performed using an effect sizes estimate (SMD) from each comparison of the control and phytase supplementation treatment for each dependent variable (*P*_SMD_ < 0.05, *I*^2^ > 50%, n ≥ 10), with phytase dosage (FTU/kg) and supplementation duration (days) as the independent (or explanatory) variable to examine the source of the meta-analysis’ detected heterogeneity.

#### 2.5.5. Descriptive Statistics

Extracted data, including the calculated chemical composition of the standalone P- and Ca-deficient basal diet, phytase dosage, and supplementation duration from the eligible studies, were analyzed using the descriptive statistics procedure of Minitab (Version 21.2, 2022).

## 3. Results

In this meta-analysis, the effects of exogenous phytase supplementation in dietary P- and Ca-deficient diet were investigated at three developmental stages (starter phase and grower and finisher phases combined) throughout the productive life span, starting with a variable P and Ca supply immediately after hatching. Broiler responses to phytase treatment were ascertained via performance, bone-breaking strength, and mineralization. 

### 3.1. Appraisal of Study and Risk Assessment for Bias

Based on the Cochrane Collaboration’s SYRCLE risk of bias tool for animal research, Figure 2 displays the risk of bias categorization for the papers included in our meta-analysis. All 41 included articles that were found to have an uncertain risk of bias for the sequence generation domain related to selection bias. However, their baseline characteristics were reported and weighed as a 100% low risk of bias. The allocation concealment of the domain of selection bias, on the other hand, constituted 2.44% for high risk and low risk of bias, respectively, while the majority (95.12%) showed an unclear risk. Regarding performance bias, all included papers appraised had a low risk of bias (100%) for both random housing and the blinding of the caregiver’s domain. In contrast, studies appraised and assessed showed almost equal proportion for low (48.78%) and unclear (43.90%) risk, with 7.32% reckoned as high risk in the case of detection bias, including random outcome assessment and the blinding of outcome assessors. Additionally, 17.07% of studies included in our meta-analysis were adjudged as unclear risk in describing the completeness of outcome data. Eligible studies of our meta-analysis were all considered free of selective outcome reporting and other problems and, therefore, appraised as a 100% low risk of bias. To summarize, out of the 41 papers that matched the eligibility criteria for our meta-analysis, 68.3% were classified as having a low risk of bias, 30% as having an uncertain risk of bias, and 1.7% as having a high risk of bias.

### 3.2. Analysis of Independent Variables Using Descriptive Statistics

Table 2 displays descriptive statistics for the P- and Ca-deficient basal diet, phytase dosage, and supplementation duration. Except for total P and Ca, which were decreased by 0.32 and 0.26 percent of DM, correspondingly, the average of the eligible papers formulated basal diets’ (negative diet) calculated chemical composition fed in d 1 to 22 or d 22 to 42 met or exceeded the NRC [2] nutrient specifications for young broilers. In addition, the average phytase dosage supplemented was 1709 (FTU per Kg of diet), with a minimum of 120 and a maximum of 40,500 in the starter phase, while the average (876) of the grower-finisher phase varied between 250 and 5000 FTU per Kg of diet. Regarding the duration of exposure to phytase, it ranged between 6 and 22 days and 10 and 33 days for the starter and grower-finisher phases, respectively.

### 3.3. Meta-Analysis

#### 3.3.1. Growth Performance

The supplemental phytase impacts on starter and grower-finisher meat-type chickens’ growth performance were summarized using random effects meta-analysis models. The standardized mean difference estimates of ADFI, ADG, BWG, and FCR in the control and phytase supplementation treatment groups with corresponding heterogeneity estimates are shown in Table 3. In the starter phase (1–22 days), supplemental phytase increased ADFI (*P*_SMD_ < 0.0001), ADG (*P*_SMD_ = 0.0006), and BWG (*P*_SMD_ < 0.0001). However, the estimate of FCR decreased with the dietary supplementation of phytase. Similarly, in the grower-finisher phase (22–42 days), supplemental phytase significantly increased ADFI (*P*_SMD_ = 0.0123) and BWG (*P*_SMD_ = 0.002) but non-significantly decreased ADG (*P*_SMD_ = 0.0699). Consistent with the supplemental phytase’s influence on broiler chicken’s FCR as indicated in the starter group meat-type chicken, it significantly reduced (*P*_SMD_ < 0.0001) in the grower-finisher phase. Supplementing phytase positively impacted growth responses in both phases, although the extent of phytase impact was much higher in the starter phase.

#### 3.3.2. Bone Strength and Mineralization

The impact of supplemental phytase on broilers’ bone strength and mineralization responses was characterized using meta-analysis’s random effects models and displayed in Table 4. The supplementation of the P- and Ca-deficient basal diet with exogenous phytase improved the BBS and tibia ash (*P*_SMD_ < 0.0001), tibia Ca (*P*_SMD_ = 0.0217), and tibia P (*P*_SMD_ = 0.0024) of broilers. Similar observations of phytase improvement for BBS and tibia ash (*P*_SMD_ < 0.0001), tibia Ca (*P*_SMD_ = 0.0047), and tibia P (*P*_SMD_ = 0.0036) were also recorded in the grower-finisher broiler. Notwithstanding supplemental phytase’s general improvement of bone strength and mineralization responses, its impact comparatively was reduced for BBS (SMD = 4.68 vs. 4.33) and tibia ash (SMD = 11.78 vs. 5.88) and heightened for tibia Ca (SMD = 4.35 vs. 4.94) and tibia P (SMD = 5.26 vs. 5.72) among broilers of the grower-finisher phase of growth.

### 3.4. Heterogeneity and Sensitivity Analysis (Publication Bias)

Table 3 and Table 4 indicate the X^2^ statistics (Q) test of heterogeneity and the corresponding proportion of total between-study variation in effect size estimates. In all the growth performance, bone strength, and mineralization responses measured in our meta-analysis, the estimates of the Q test revealed statistics with *p*-values ≤ 0.10, which were considered significant between-study variability with their corresponding proportions of heterogeneity (*I*^2^) ranging between 93.3 and 99.8%. With reference to the Borenstein et al. [7] benchmark range for heterogeneity, the *I*^2^ statistics detected among analyzed outcomes in our MA were interpreted as representing very high levels of heterogeneity, as all the proportions fell within 75% ≤ *I*^2^ ≤ 100%. To address this concern, it was imperative to explore what causes this heterogeneity.

On the other hand, visualized funnel plots (Appendix A) and the *p*-values of all the measured outcomes in the MA using Egger’s regression asymmetry test revealed a significant bias (*p* ≤ 0.05). With sufficient evidence of publication bias detected among all the outcome variables in the MA, the results of the corrected publication bias by Duval and Tweedie’s “trim-and-fill” method (illustrated in Table 5) suggest that improvements seen in terms of the effect size estimate of growth performance, bone strength, and mineralization outcomes are actual effects of the supplemental phytase. Nonetheless, the magnitude of the effect sizes on measured responses in our MA, except BWG, were non-significantly reduced in our adjustment of publication bias (Table 5) compared to the observations in Table 3 and Table 4 in both phases of development.

### 3.5. Subgroup Analysis of Broiler Strain and Basal Diet’s Dietary P Source Association with Phytase Efficacy on Growth Performance, Bone Strength, and Mineralization

Subgroup analysis was performed to test the hypothesis that supplementing phytase in a P- and Ca-deficient diet is more effective among some strains of broilers as well as basal diets’ dietary P sources than others (i.e., the studies included in our meta-analysis are not drawn from a single overall population), supposing that they belong to several subgroups and that each subgroup has a unique overall impact [87]. Furthermore, in the case of a few studies within subgroups (say, five or fewer), estimates within subgroups are generally likely to be subject to greater sampling variation and, hence, are less precise [88,89]. Therefore, subgroups with just NC = 1 were not reported in our meta-analysis. 

#### 3.5.1. Broiler Strain and Growth Performance

The summary effects of the response of ADFI, ADG, BWG, and FCR, as illustrated in Table 6, revealed that the efficacy of phytase in a Ca- and P-deficient diet generally varied significantly (*p* < 0.0001) among broiler strain subgroups in both phases of production.

Among subgroups of starter broiler strains, supplementation with phytase consistently increased (*p* < 0.0001) the effects on ADFI, which was shown to be more pronounced and precise among some strains of broilers than others (Cobb 500 > Ross 308 > Ross > Ross × Ross 708 > Ross 708). Phytase’s positive effect on feed intake predictably translated to an improved (*p* < 0.0001) response of broiler’s ADG, with much of it seen in Ross 308 compared to Cobb 500 strains. However, supplementation with phytase reduced (*p* < 0.0001) ADG in Ross strains. Interestingly, all subgroups of broiler strains recorded increased (*p* < 0.0001) BWG, with the observed effects pronounced among strains, including Cobb 500, Ross, Ross 708, Ross 308, Yield Plus × Ross 708, and Ross × Ross 708. Contrariwise to the effects of phytase intervention on starter broilers’ ADFI, ADG, and BWG, broilers’ FCR was significantly (*p* < 0.0001) decreased among Ross, Ross 708, Cobb 500, Ross 308, Hubbard × Cobb 500, Yield Plus × Ross 708, Ross × Ross 708, and Lohman Indian RSR, except in the Ross 708 × YPM strain that recorded an increase. Similarly, among the growing-finishing broiler subgroups, ADFI (*p* = 0.0044), ADG (*p* < 0.0001), and BWG (*p* < 0.0001) indicators of growth were heightened in Cob 500 and Ross 308 strains, except the strain of Ross that recorded reduced effects. In contrast, a decreasing trend in FCR (*p* < 0.0001) was observed in Cobb 500, Ross 308, and Ross strains when fed P- and Ca-deficient basal diet supplemented with phytase.

It was observed that the impact of supplemented phytase in ameliorating the effects of the P and Ca deficiency on growth performance seemed to increase (ADFI, ADG, and BWG) and reduce (FCR) at each time point in both the starter and grower-finisher phases. However, the magnitude of phytase effects was pronounced and more precise in some strains of broilers than in others, generally during the starter phase than the grower-finisher phase.

#### 3.5.2. Broiler Strain and Bone Strength and Mineralization

The subgroup analyses of broiler strains’ correlation with phytase effects on bone strength and mineralization in starter and grower-finisher phases are displayed in Table 7. Supplementing exogenous phytase in a low Ca and P-diet positively (*p* < 0.05) influenced bone mineralization indicators, including tibia ash, tibia Ca, and tibia phosphorus, but not bone-breaking strength in both phases of growth in our meta-analysis.

The bone-breaking strength (BBS) of a broiler chicken supplemented with phytase at three weeks (*p* = 0.4776) and six weeks (*p* = 0.19) was not significantly associated with broiler strain, suggesting that effect sizes are consistent across subgroups. Yet, Ross 308, Cobb 500, and Ross strains’ tibia ash, calcium, and phosphorus were pronouncedly influenced via phytase supplementation in low P and Ca diets in both phases in our meta-analysis. Relatively, the magnitude of the effects of phytase varied with broiler strain and growth phase. Whereas BBS had reduced in both Cobb 500 and Ross 308 in the grower-finisher phase, the impact of supplemental phytase increased all bone mineralization indicators assessed in Cobb 500 strains. However, it yielded lower SMDs in Ross 308 in the grower-finisher phase. 

#### 3.5.3. Basal Diets’ Dietary P Source and Growth Performance

Statistics from subgroup analyses of basal diet’s dietary P as a moderator of phytase effects on responses of growth in broilers in both developmental stages in our meta-analysis are presented in Table 8. At three weeks, Table 8 shows that broilers’ ADFI (*p* = 0.0024), ADG (*p* < 0.0001), and FCR (*p* < 0.0001) varied significantly. Yet, there was no dietary P source influence on broilers’ BWG (*p* = 0.3167). Except for WRES (SMD = −0.32), which decreased, supplemental phytase effects on broiler’s ADFI increased when fed corn–soybean (SMD = 15.33), wheat–soybean (SMD = 10.10), corn–soybean–DDGS (SMD = 8.99), CSRRB (SMD = 6.56), and wheat–corn–soybean (SMD = 1.86). Similarly, ADG increased when broilers were fed corn–soybean (SMD = 9.31) and wheat–soybean (SMD = 23.02) but reduced with wheat–corn–soybean (SMD = −0.31) being fed. Body weight gain was not associated substantially with the basal diet’s dietary P source, signifying a consistent effect of phytase across the subgroups. In contrast, a general reducing effect of supplemental phytase on FCR is seen in all dietary P source subgroups, with a substantial impact observed when corn–soybean (SMD = 10.79) and wheat–corn–soybean (SMD = 4.35) were fed as basal diets.

Unlike the response pattern at three weeks, dietary P source as a moderator in our subgroup analysis significantly influenced ADFI (*p* < 0.0001), ADG (*p* = 0.0059), BWG (*p* = 0.0002), and FCR (*p* = 0.0005) at week 6 of growth. Supplemental phytase improved the ADFI of broilers fed corn–soybean (SMD = 86.28), corn–soybean–DDGS (SMD = 39.35), and WRES (SMD = 0.62) but reduced intake when supplemented to wheat–corn–soybean (SMD = −1.10). Comparable observations in the effects of phytase were revealed in the ADG and BWG of broilers fed wheat–corn–soybean but improved when supplemented to corn–soybean, corn–soybean–DDGS, and WRES. On the contrary, FCR decreased when phytase was augmented in all basal diets, with highly significant effects observed in corn–soybean and WRES. In general, feeding starter and grower-finisher broilers with a corn–soybean diet supplemented with phytase was much more beneficial and consistent in improving the growth performance of the broiler.

#### 3.5.4. Basal Diets’ Dietary P Source and Bone Strength and Mineralization

The bone strength and mineralization of broilers supplemented with phytase are illustrated in Table 9. The basal diet’s dietary P source was a significant predictor for BBS (*p* = 0.0330), tibia ash (*p* < 0.0001), and tibia Ca (*p* = 0.5041) of broilers of 3 weeks, but not tibia P (*p* = 0.1534). Except in tibia Ca, the results indicated consistent effects necessitated by phytase in broilers fed corn–soybean, wheat–soybean, and wheat–corn–soybean diets. However, wheat-based diets realized a much greater effect size than corn-based diets in all responses. Like the starter phase, the effect size of BBS (*p* < 0.0001), tibia ash (*p* = 0.0007), tibia Ca (*p* < 0.0001), and tibia P (*p* < 0.0001) of broilers in week six was significantly predicted by the sources of P fed in our subgroup analysis. Although the corn–soybean basal diet was associated with better effects of phytase in measured outcomes compared to wheat–corn–soybean, greater bone strength and mineralization are generally more associated with a wheat–soybean and WRES-based diet.

### 3.6. Effects of Phytase Dosage and Supplementation Duration on Mineralization, Bone Strength, and Growth Performance in Broilers

A meta-regression analysis was carried out to investigate phytase dose–response and exposure duration to identify the summary trend emerging from multiple studies answering our research question. Analyses were performed to test whether there is any association, whether the response changes approximately at a constant rate throughout the observed outcome effects, and whether there is any substantial change in the outcome beyond the average treatment effects. 

#### 3.6.1. Meta-Regression Analysis of Phytase Dose Response

The dose–response meta-analysis using regression models of broiler growth, bone strength, and mineralization per developmental phase are summarized in Table 10. During the starter phase (at three weeks), the covariate “phytase dosage” had no significant relationship (*p* > 0.05) with ADFI, ADG, BWG, or FCR. Likewise, the dosage of phytase supplemented to starter broilers had no significant relationship (*p* > 0.05) with BBS, tibia ash, and tibia calcium, except for tibia P, which was significantly changed (intercept = 9.5629; estimate = −0.0032; *p* = 0.0467) with a unit increase in dosage, explaining 15.50% of the observed heterogeneity for tibia P. The bubble plot (Figure 3), which demonstrated a considerable detrimental effect on tibia P owing to increased phytase dose (500–3000 FTU per kg of diet), emphasizes this observation.

On the other hand, the results of the meta-regression models during the grower-finisher phase indicated a non-significant (*p* > 0.05) phytase dosage effect for growth outcomes (ADFI, ADG, BWG, and FCR) and bone strength and mineralization (BBS, tibia ash, tibia Ca, and tibia P).

#### 3.6.2. Phytase Exposure Duration

Table 11 shows a summarized effect size investigating the association between phytase exposure duration with growth performance, bone strength, and mineralization arising from both developmental phases of broilers in the eligible studies of our MA. The results of the starter phase indicated that there was a non-significant (*p* > 0.05) association between phytase exposure duration and growth outcomes measured, except for FCR (intercept = −29.0970; estimate = 1.1359; *p* = 0.0132), which improved significantly per unit increase in phytase duration. Equally, phytase exposure for a longer duration in broilers in the early phase of their development did not correlate significantly with bone-breaking strength and tibia ash but substantially lowered the effect size of tibia Ca (intercept = 19.4068; estimate = −0.9962; *p* = 0.0074) and tibia P (intercept = 24.2718; estimate = −1.2321; *p* < 0.0001). Phytase exposure duration as a predicting variable in our meta-regression models further explained 29.05% and 58% of the observed heterogeneity in tibia Ca and P accordingly. 

On the other hand, in grower-finisher broilers, the duration of phytase exposure was a non-significant (*p* > 0.05) contributing predictor of the association effects in all growth-measured outcomes of broilers-supplemented phytase. However, every unit change in the duration of supplementation increased the association’s effect size of tibia Ca (intercept = −43.7178; estimate = 2.5215; *p* = 0.0034) and tibia P (intercept = −56.0797; estimate = 3.2005; *p* = 0.0001) significantly, explaining 40.79% and 56.07% of the heterogeneity in our global study. Remarkable improvements in tibia mineralization are seen during the late phase of broiler development as compared to the starter phase, which reveals increasing effects, as shown in the bubble plots (Figure 4). Notwithstanding these observations in tibia mineralization, supplementation duration did not substantially (*p* > 0.05) impact the effect size of bone-breaking strength and tibia ash. 

## 4. Discussion

Phytase activity, defined in *fytase* units (FTU), is the amount of enzyme that liberates 1 μmol of inorganic orthophosphate/min from 0.0051 molL^−1^ sodium phytate at pH 5.5 and at a temperature of 37 °C [90]. As documented in numerous studies, data on bone integrity and growth performance are often used to estimate and validate phytate P liberation by phytase, as these parameters provide additional useful baseline information [32]. More precisely, the supplemental phytase effects on broiler chicken growth performance, bone strength, and mineralization have been observed in P- and Ca-inadequate diets with variable influence. To ascertain exogenous phytase global effects, meta-analyses focused on non-phytate phosphorus, phosphorus retention, and interactive effects of Ca, vitamin D3, and *Eimeria* infections have previously been explored [42,43,44,45,46]. Yet, as a broiler genetic framework, new generations of EP, and new research continue to proliferate in this research domain, it is imperative to evaluate P utilization in current broiler strains, dietary P sources, and physiological stages. 

### 4.1. Appraisal of Study and Assessment of Risk of Bias

A systematic review’s quality of evidence is as important as analyzing the data [87]. In fact, the credibility of the data and the results of the individual studies included in the systematic review are closely related to the quality and reliability of the evidence [91]. Therefore, measuring the quality of a particular study by assessing its risk of bias is fundamental in systematically reviewing a research question. Spanning between 2006 and 2024, the studies integrated into our review tail towards the period of high demand for the adoption and use of similar reporting standards in methodological descriptions of animal intervention data-driven research. This notwithstanding, 30% of the review’s included studies were prone to an unclear risk of bias. The magnitude of unclear bias, significantly introduced by a lack of blinding, allocation concealment, sequence, and inadequate randomization in our appraisal, ascertains that similar reporting standards are not yet universal in animal intervention experimentation reporting. Our observation substantiates the arguments of Macleod et al. [91] and Kilkenny et al. [92], who hold the view that most individual research fails to report blinding, allocation concealment, sequence generation, and randomization, which are not standard practices in animal experiment reporting. In support of our findings, Nuamah et al. [87] and Ncho et al. [93], both in a meta-analysis of animal intervention studies, found these risk domains as the main contributors of unclear bias. However, with an overall low risk of bias (68.3%) and just 1.7% high risk of bias, the evidence of our meta-analysis is credible and valid in measuring the overall effect size of phytase supplemented with inadequate P- and Ca-basal diets in broilers. 

### 4.2. Impact of Exogenous Phytase on Growth Performance

As a general concept, the addition of phytase is well documented to hydrolyze phytate and reduce its anti-nutritional effects, thereby offering an improved availability and digestibility of P, which have consistently been shown to support and enhance broiler performance and nutrient utilization when fed with P-deficient diets [11,23,94,95,96,97]. Our meta-analysis examined the growth performance of broilers from 1 to 42 days post-hatch. In the two 3-week periods of d 1 to 22 and d 22 to 42, performance indices, including ADFI and BWG, increased, while FCR reduced during both periods, especially with a higher magnitude of phytase effect size in d 1 to 22 broilers fed the corn–soybean dietary P source diet. This suggests that at all growth phases, phytase facilitates the bioavailability of enough phytate P from a P- and Ca-deficient diet to meet the requirement for growth.

Exogenous enzymes, including phytase influence on performance, generally correlate with increases in nutrient digestibility [83,94,98], as it is renowned for breaking phytate complexes to liberate P and other essential nutrients in the diet, thereby increasing productive performance in birds with diets minimal in avP [99] or both Ca and avP [13]. Thus, the current global study confirms, as pointed out in the meta-analyses of Letourneau-Montminy et al. [45], Kermani et al. [44], and Shi et al. [78], whose studies illustrated the positive impact of phytase on P utilization in broilers through improvement in productive performance, particularly BWG and FCR, with the most remarkable improvements realized in P- and Ca-deficient diets [100,101,102]. This assertion, according to Tamim et al. [103] and Gifford and Clydesdale [104], is accounted for by the less insoluble Ca-phytate complex formation in the gut owing to the lower Ca level in the diet and the lower pH of the gut, which maintains a modest Ca-P balance in the diet. The poultry utilization of phytate P is inclined to the dietary Ca and P content [104,105,106] since Ca could quickly form a precipitate with phytate in the gut [107] and reduce the solubility of InsP_6_, making it less accessible to phosphatases [23]. For optimal broiler growth, feeding inadequate diets of P and Ca, the calcium concentration in the diet must essentially be adjusted to a level that optimizes P utilization. 

Regarding feed intake, broilers’ overall phytase effect size (Table 3) supports the claims that incremental phosphorus release is linked with feed intake increase, as P often acts as a regulator of feed intake [108,109]. Synergistically with other enzymes, supplemental phytase in poultry increases the transit rate through the gut, thus increasing feed intake [110]. Furthermore, the improved feed intake might also be due to the change in the viscosity of the diets [111], possibly attributed to the heightened P availability, other minerals, and nutrients, facilitating a higher-quality diet [112].

Supplemental phytase effect size in BWG and FCR in the present meta-analysis may be ascribed to an improvement in the P- and Ca-inadequate diet’s energy value, necessitated by the activity of the liberated phytate P. The improvement in BWG and FCR is possibly linked to the bird’s ability to utilize the energy of feed metabolize in the body owing to the diet’s increase in protein and amino acid digestibility, which correlates positively with growth performance indices [113]. It has been suggested by Humer et al. [114] that the mechanism of energy improvements, particularly with phytase association, may be due to the dissolution of the phytate complexes, which promote protein absorption and increase carbohydrate and fat digestibility. Partly related to the improvement could be the exogenous phytase’s ability to offset endogenous losses to enhance metabolic energy by lowering the energy required for maintenance, thus granting a significant amount of energy for growth. Further confirming our observations, the work of Abd El-Hack et al. [115] ascribed the superior broiler performance associated with phytase supplementation to an increase in energy liberated from the diet owing to a boost in nutrient digestibility via the release of nutrients bound to phytic acid and increased phosphorus utilization efficiency. 

Supporting the liberation of excess P, the improvement in growth performance may be ascribed to the excess phytase dose (Table 2) supplemented in the starter phase. Thus, the superdosing of phytase beyond the commercial recommendations will result in additional phytate hydrolysis, leading to improved BWG and FCR, particularly in young broilers [13]. Likewise, the extra-phosphoric effect of the phytase, arising from the liberation of *myo*-inositol, the final product of phytate dephosphorylation, might have partly contributed to growth improvement in both phases. Specifically, *myo*-inositol generated via the dephosphorylation of the whole phytate increases the growth rate and feed efficiency of broiler chickens because of its crucial role in cell growth and metabolism, fat deposition and transport, protein deposition and transport, and gluconeogenesis [116,117].

However, improvements in broiler performance via the phytase supplementation of P- and Ca-inadequate diets are not solely mediated by the exogenous phytase but also depend on the feed composition, mineral content, source of the phytase, species and age of birds, and the endogenous microbiota that affects the pH range in the gut. [115,118]. In the present meta-analysis, although exogenous phytase’s impact to ameliorate the effects of the P and Ca deficiency on growth performance appeared to increase ADFI, ADG, and BWG and reduce FCR at each phase, the extent of its impact is strain (Table 7), age, and source-dependent dietary P (Table 8), with a much higher impact observed in the starter phase. The findings of the present study contradict those of Edwards Jr et al. [119], who reported that the ability to utilize phytate P increases with the age of poultry, coupled with Peeler [120], who, in an earlier review, stated that the bioavailability of phytate phosphorus is moderate for adult poultry but very low for young poultry. Converse to these assertions, a recent study by Abudabos [121] confirms our observation regarding phytate P utilization with age, affirming that the demand for P is more pronounced in young poultry for body weight gain. Moreover, the utilization of nutrients in chicks is more pronounced during the first two weeks because the rapid development and growth of organs and tissues occur during this period [29,30]. In the case of broiler strain, a significant prediction of growth performance in the current study inclines to the statement that the genetics of chicken influence the magnitude of indices of growth [122]. Cobb 500 and Ross 308 strains probably had a higher adaptive capacity to the deficient diet, and so increased their intestinal phytase and phosphatase activities, which generally improved performance and mineralization, touting them as efficient nutrient utilizers of low P and Ca diets supplemented with phytase. 

Interestingly, the magnitude of growth responses throughout the production cycle of birds appears to also depend on the dietary P source, i.e., type of ingredient (Table 8) and supplementation duration (Table 11) other than the dosage of phytase (Table 10), as both explain a significant portion of the heterogeneity in our global study. This suggests that phytase activity differs per the dietary P source as each source shares distinct phytate susceptibility, probably because of their structure and storage of phytate. Consistent with Ravindran et al. [123], the difference is predominantly due to the solubility of the dietary P source at acidic pH, as phytase is more readily hydrolyzed by soluble phytate. Accordingly, the magnitude of the response to phytase is directly proportional to the concentration of phytate in ingredients, with poorly digestible feed ingredients showing a greater response to phytase than those with higher inherent digestibility. The discrepancy between the corn–SBM-based diet and wheat–SBM-based diet in our subgroup analysis may partly be associated with the propensity and storage site of phytate in the grains [124], which is more likely to complex wheat protein than that of maize [111]. Our observation was in line with Selle and Ravindran [23], in that phytase hydrolyzes phytate in soybean meal and maize more readily than wheat. 

Even though phytase in broiler diets has been supplemented until market weight, the findings from the present study suggest that the efficacy of phytase on growth response is more tailored to the duration of its supplementation and not the dosage. This supports the theory that more extended feeding periods may acclimatize broilers to low-P diets to maintain P homeostasis adaptations for the digestive ability of the birds, resulting in a higher digestibility of P [125,126,127]. 

### 4.3. Impact of Exogenous Phytase on Bone Strength and Mineralization

Bone ash, a highly sensitive metric used for dietary P availability evaluation [128], alongside bone mineralization, an efficient parameter for the prediction of the quality of the bone attributable to inorganic P release by phytase [32,129], are both recognized for their beneficial effects associated with broiler muscular development and performance. Furthermore, both bone parameters tend to be more precise and sensitive indicators of changes in P and Ca availability than vital growth indices [107,130,131,132]. 

In this study, the low P and Ca diet supplemented with phytase increased bone strength and tibia ash, thus predictably increasing bone mineralization at each age. However, the magnitude of BBS and tibia ash decreased, whereas tibia Ca and P increased at d 42. Tibia ash is known to reduce the sensitivity of the bones to dietary changes as the bird ages [133]. The supplementation of phytase seems to be more influential in increasing tibia ash during the early phase of growth of broilers, confirming the evidence of Talaty et al. [134] that bone mineral density peaked at four weeks of age. According to the authors, the tibia continues to grow, especially after the 3 to 4 weeks growth spurt, but does not become denser in ash because of the increase in surface area. The observations of the present study are close to those reported in preceding studies [3,8,18,94], which implies that phytase can alleviate the effects of phosphorus- and calcium-availability reduction while maintaining the similar bone ash, bone mineralization, and bone-breaking strength of broilers. These observations may be due to the role of phytase and P, with phytase liberating phytate P in diets low in P and Ca. Almost 85% of P is retained in bone, yet Ca provision is required for P deposition through the formation of hydroxyapatite, a mineral phase of the bones [135]. With calcium accentuated as an essential nutritional factor in upholding bone strength, bone strength and mineralization will be reduced without a sufficient dietary calcium source due to blood hypocalcemia [136]. Hence, a marginal P and Ca supply in broiler diets will adversely impact bone formation and mineralization, except in phytase supplementation [137]. Likewise, the release of additional bone-forming compounds, including zinc, amino acids, and others, also improved bone integrity [138].

Nonetheless, the outcomes of tibia P and Ca and tibia ash (to some extent) were not alone influenced by the phytase supplied but mainly by its duration of feeding (Table 11), dietary sources of P (Table 9), and strains of broilers (Table 7) in this study. This may be due to the increased negative impact of the inadequate diet on tibia ash over longer durations than shorter feeding durations. Thus, there is an opportunity for continued P and Ca deposition on the bone during this phase of development [94] after prolonged exposure to the ameliorative effects of phytase in diets low in P and Ca. Our findings support the assertion that bone integrity is influenced by rapid growth, genetics, the environment, management, nutrition, and age [136].

## 5. Conclusions

The systematic review using meta-analysis models indicates that the supplementation of broiler P- and Ca-deficient basal diets with phytase improved ADFI, BWG, and FCR, especially when added to starter diets. Moreover, exogenous phytase supplementation supports bone strength and mineralization in broilers fed P- and Ca-deficient diets at each time of growth. However, the effects of bone ash and strength can be maximized in the starter phase of development. Despite phytase superiority, the effect sizes of outcomes evaluated were related to the age and strain of the broiler, dietary P source, and duration of phytase exposure, rather than the dosage. Feeding broilers, especially Cobb 500 and Ross 308 strains, with a corn–soybean diet supplemented with phytase was much more beneficial and consistent in improving the growth performance, whereas wheat–soybean and WRES-based diets were associated with the superior mineralization and strength of the bones. 

## Figures and Tables

**Figure 1 animals-14-02090-f001:**
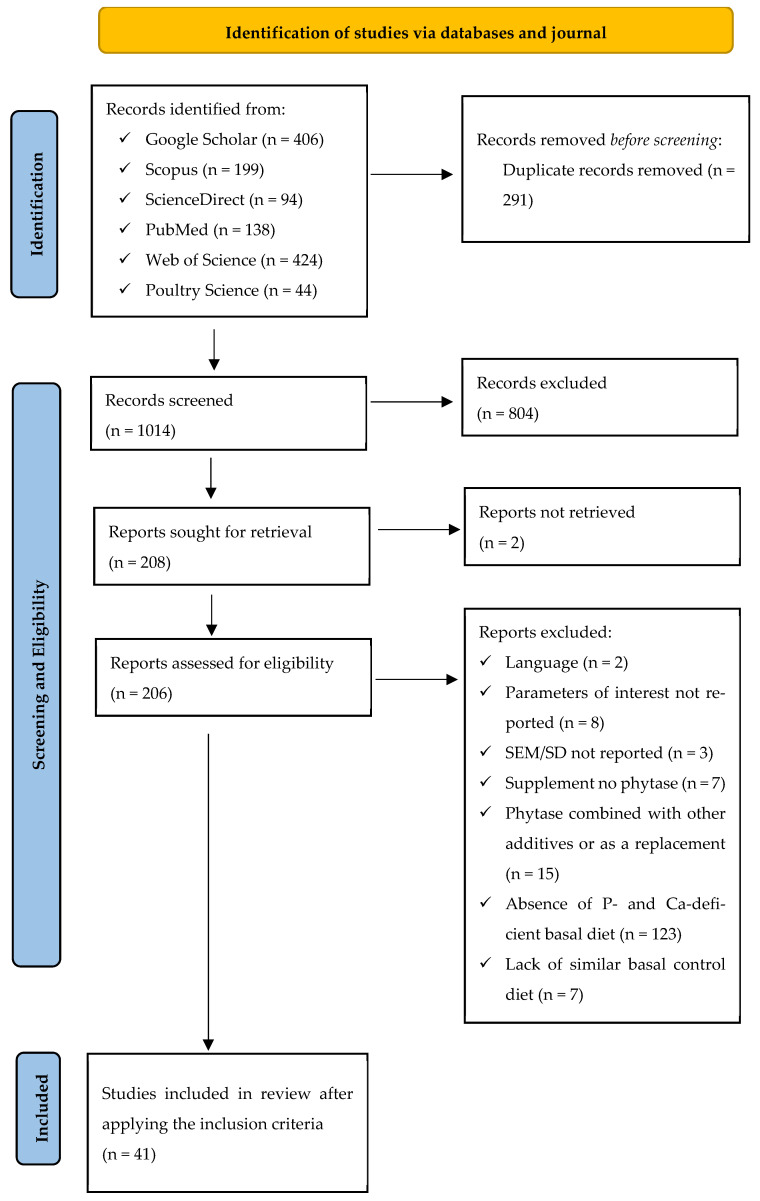
The PRISMA diagram detailing the systematic literature search and paper selection process.

**Figure 2 animals-14-02090-f002:**
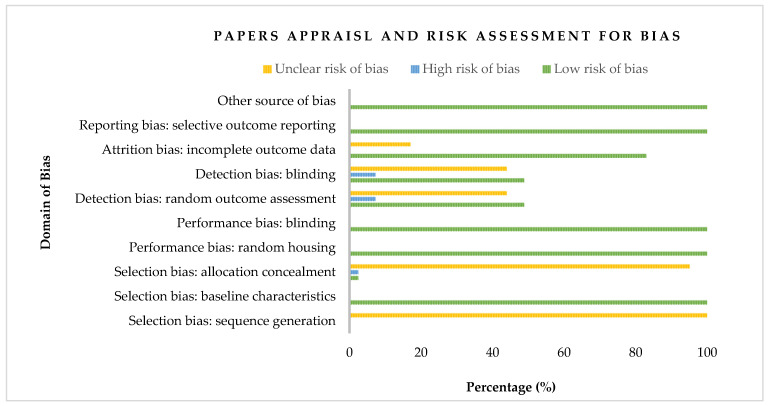
A bar graph showing the risk of bias classification of eligible papers.

**Figure 3 animals-14-02090-f003:**
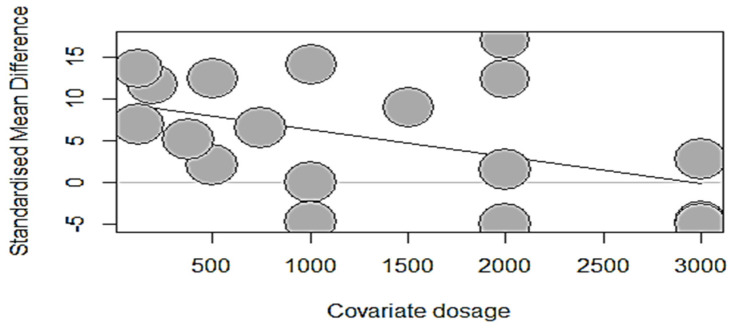
A decreasing trend in tibia P’s standardized mean difference per unit increase in phytase dosage.

**Figure 4 animals-14-02090-f004:**
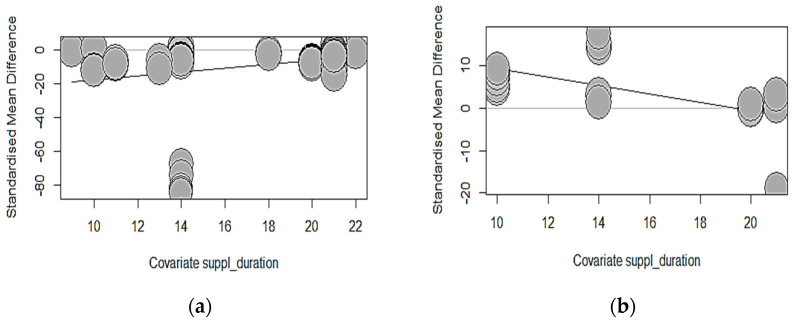
Showing bubble plot of standardized mean difference in (**a**) starter FCR; (**b**) starter tibia Ca; (**c**) starter tibia P; (**d**) grower-finisher tibia Ca; (**e**) grower-finisher tibia P associated with a unit change in duration of phytase exposure.

**Table 2 animals-14-02090-t002:** Descriptive statistics of the basal diet’s calculated composition, phytase dosage, and supplementation duration.

	Starter Phase (1–22 Days)				Grower-Finisher Phase (22–42)			
Variable	Mean	SEM	Minimum	Maximum	Mean	SEM	Minimum	Maximum
Crude Protein, % of DM	21.91	0.22	19.40	26.33	20.48	0.47	17.80	23.00
Metabolizable energy (MJ/kg)	12.44	0.24	11.97	14.10	12.82	0.15	12.18	13.34
Digestible Met + Cys, % of DM	0.92	0.02	0.77	1.19	0.80	0.05	0.65	0.86
Digestible Lysine, % of DM	1.11	0.05	0.10	1.42	0.81	0.19	0.10	1.19
Digestible Thr, % of DM	0.78	0.01	0.63	0.89	0.71	0.03	0.6175	0.76
Digestible Val, % of DM	0.93	0.04	0.77	1.30	0.86	0.10	0.76	1.05
Available Phosphorus, % of DM	0.26	0.01	0.10	0.35	0.21	0.03	0.17	0.29
Total P, % of DM	0.53	0.01	0.36	0.65	0.47	0.04	0.41	0.60
Calcium, % of DM	0.74	0.02	0.40	0.95	0.64	0.04	0.47	0.74
NPP, % of DM	0.30	0.02	0.20	0.43	0.30	0.04	0.21	0.42
Phytase dosage (FTU per Kg of diet)	1709	397	120	40,500	876	153	250	5000
Supplementation Duration, day	16	0.68	6.00	22.00	20	1.48	10.00	33.00

**Table 3 animals-14-02090-t003:** Growth performance of starter and grower meat-type chickens supplemented with phytase.

		Random Effects Model			Heterogeneity				Egger’s Test ^b^
Parameter	n(NC)	SMD	95% CI	*p*-Value	Chi-Squared (Q)	*I*^2^ (%)	τ^2^	*p*-Value ^a^	*p*-Value
*Starter Phase (1–22 days)*									
ADFI, g/day	38(97)	12.57	6.61, 18.54	<0.0001	25,654.46	99.6	895.47	0.000	<0.0001
ADG, g/day	6(13)	11.31	4.85, 17.76	0.0006	22,789.26	99.8	140.72	0.000	<0.0001
BWG, g	38(96)	10.76	8.13, 13.78	<0.0001	22,789.26	99.6	171.11	0.000	<0.0001
FCR, g/g	33(90)	−8.99	−12.59, −5.39	<0.0001	18,235.57	99.5	300.91	0.000	<0.0001
*Grower-Finisher Phase (22–42 days)*									
ADFI, g/day	11(26)	47.35	10.27, 84.42	0.0123	7353.11	99.7	9285.23	0.000	<0.0001
ADG, g/day	3(9)	7.78	−0.63, 16.19	0.0699	2676.45	99.7	165.52	0.000	<0.0001
BWG, g	11(22)	7.67	3.62, 11.73	0.0002	5297.35	99.6	93.53	0.000	0.0118
FCR, g/g	11(24)	−4.30	−6.38, −2.21	<0.0001	4387.11	99.5	26.65	0.000	0.0030

n: number of studies; NC: the number of comparisons between exogenous phytase treatment and deficient P and Ca control group; SMD: standardized mean difference; CI: confidence interval; Q: X^2^ statistics; *I*^2^: proportion of total variation in effect size estimate due to heterogeneity; τ^2^: heterogeneity variance of the actual effect. (Forest plots showing mean difference effects are presented in Appendix A). ^a^ *p*-value to X^2^ statistics (Q) test of heterogeneity. ^b^ Egger’s regression asymmetry test.

**Table 4 animals-14-02090-t004:** Bone strength and mineralization of starter and grower meat-type chickens supplemented with phytase.

		Random Effects Model			Heterogeneity				Egger’s Test ^b^
Parameter	n(NC)	SMD	95% CI	*p*-Value	Chi-Squared (Q)	*I*^2^ (%)	τ^2^	*p*-Value ^a^	*p*-Value
*Starter Phase (1–22 days)*									
BBS, Kgf/mm	4(17)	4.68	2.81, 6.54	<0.0001	225.00	92.9	14.41	<0.0001	<0.0001
Tibia Ash, g/Kg DM	16(51)	11.78	8.65, 14.92	<0.0001	7144.72	99.3	125.08	0.000	<0.0001
Tibia Ca, g/Kg DM	4(17)	4.35	0.64, 8.06	0.0217	550.00	97.1	59.77	<0.0001	0.0030
Tibia P, g/Kg DM	4(18)	5.26	1.87, 8.65	0.0024	862.39	98.0	52.72	<0.0001	0.0089
*Grower-Finisher Phase (22–42 days)*									
BBS, Kgf/mm	3(11)	4.33	2.59, 6.08	<0.0001	191.72	94.8	7.92	<0.0001	0.0013
Tibia Ash, g/Kg DM	6(21)	5.88	3.13, 8.62	<0.0001	716.64	97.2	39.13	<0.0001	0.0005
Tibia Ca, g/Kg DM	4(13)	4.94	1.51, 8.36	0.0047	179.30	93.3	37.63	<0.0001	<0.0001
Tibia P, g/Kg DM	4(13)	5.72	1.86, 9.57	0.0036	229.97	94.8	47.75	<0.0001	0.0002

n: number of studies; NC: the number of comparisons between exogenous phytase treatment and deficient P and Ca control group; SMD: standardized mean difference; CI: confidence interval; Q: X^2^ statistics; *I*^2^: proportion of total variation in effect size estimate due to heterogeneity; τ^2^: heterogeneity variance of the actual effect; BBS: bone-breaking strength. (Forest plots showing mean difference effects are presented in Appendix A). ^a^ *p*-value to X^2^ statistics (Q) test of heterogeneity. ^b^ Egger’s regression asymmetry test.

**Table 5 animals-14-02090-t005:** Corrected publication bias of phytase effect on broilers’ performance, bone strength, and mineralization.

		Random Effects Model			
Parameter	NC	d.f.	SMD ^b^	95% CI	*p*-Value ^c^
*Starter Phase (1–22 days)*					
ADFI, g/d	131	130	1.6779	−5.04, 8.40	0.6244
ADG, g/d	18	17	1.8716	−6.79, 10.53	0.6718
BWG, g	137	136	3.3592	0.31, 6.41	0.0307
FCR, g/g	124	123	−2.0180	−6.15, 2.11	0.3383
BBS, Kgf/mm	24	23	1.8186	−0.59, 4.23	0.1387
Tibia Ash, g/Kg DM	74	73	2.0083	−2.63, 6.65	0.3959
Tibia Ca, g/Kg DM	22	21	1.0343	−2.93, 5.00	0.6093
Tibia P, g/Kg DM	24	23	0.9147	−3.12, 4.95	0.6568
*Grower-Finisher Phase (22–42 days)*					
ADFI, g/d	35	34	1.1746	−42.28, 44.62	0.9577
ADG, g/d	11	10	1.4517	−9.31, 12.22	0.7916
BWG, g	29	28	2.0851	−2.88, 7.05	0.4104
FCR, g/g	30	29	−1.6657	−4.26, 0.92	−1.6657
BBS, Kgf/mm	16	15	2.2862	0.19, 4.38	0.0324
Tibia Ash, g/Kg DM	28	27	2.1719	−1.74, 6.08	0.2761
Tibia Ca, g/Kg DM	16	15	1.7751	−2.89, 6.44	0.4555
Tibia P, g/Kg DM	18	17	0.6856	−4.28, 5.65	0.7868

NC: the number of comparisons between exogenous phytase treatment and deficient P and Ca control group; CI: confidence interval; d.f: degree of freedom. ^b^ standardized mean difference after adjustment for funnel plot asymmetry using the trim-and-fill method (random effects). ^c^
*p*-value of the effect estimate after adjustment for funnel plot asymmetry using the trim-and-fill method (random effects).

**Table 6 animals-14-02090-t006:** Meta-ANOVA of broiler strain association with measured growth outcomes.

			Random Effects Model					
Parameter	Covariate: Strain	NC	*I*^2^, %	SMD (95% CI)	τ^2^	d.f	Q	*p*-Value ^a^
	*Starter Phase (1–22 days)*							
ADFI, g/d	Ross 308	41	99.4	5.61 (3.67, 7.55)	39.71	10	737.59	<0.0001
	Ross 708	6	98.8	1.35 (0.60, 2.09)	0.86			
	Cobb 500	19	99.6	11.54 (5.09, 18.00)	205.46			
	Hubbard × Cobb 500	4	83.4	151.73 (138.71, 164.75)	146.01			
	Ross	10	99.4	1.83 (−1.36, 5.02)	26.32			
	Ross × Ross 708	5	99.6	1.70 (−1.51, 4.92)	13.45			
	Ross 708 × YPM	3	98.8	35.12 (23.05, 47.31)	113.55			
	Yield Plus × Ross 708	5	99.6	2.56 (0.90, 4.23)	3.60			
	Lohman Indian RSR	2	96.1	11.54 (9.48, 13.60)	2.12			
ADG, g/d	Ross 308	6	99.9	15.52 (5.14, 25.91)	168.09	3	264.88	<0.0001
	Cobb 500	3	89.6	18.65 (16.23, 21.06)	4.10			
	Ross	3	99.4	−0.31 (−3.06, 2.43)	5.85			
BWG, g	Ross 308	43	99.4	7.67 (5.26, 9.72)	46.42	10	1122.71	<0.0001
	Ross 708	6	96.0	7.69 (6.56, 8.81)	1.90			
	Cobb 500	19	99.6	19.36 (8.12, 30.60)	624.18			
	Hubbard × Cobb 500	4	83.4	20.70 (18.89, 22.52)	2.86			
	Ross	7	99.0	16.13 (11.23, 21.03)	42.04			
	Ross × Ross 708	5	99.8	5.49 (−1.79, 12.77)	68.80			
	Ross 708 × YPM	3	97.4	2.81 (1.90, 3.73)	0.63			
	Yield Plus × Ross 708	5	99.4	5.58 (3.42, 7.74)	6.02			
	Lohman Indian RSR	2	94.3	13.41 (11.31, 15.50)	2.16			
FCR, g/g	Ross 308	38	99.1	−4.29 (−5.38, −3.21)	11.37	10	1434.92	<0.0001
	Ross 708	6	82.8	−7.68 (−8.22, −7.14)	0.39			
	Cobb 500	15	99.4	−4.84 (−6.92, −2.75)	16.97			
	Hubbard × Cobb 500	4	82.2	−7.18 (−7.83, −6.53)	0.37			
	Ross	10	99.6	−48.69 (−72.95, −24.43)	1511.60			
	Ross × Ross 708	5	99.6	−2.66 (−5.84, 0.53)	13.20			
	Ross 708 × YPM	3	99.2	0.37 (−0.90, 1.65)	1.26			
	Yield Plus × Ross 708	5	98.0	−4.06 (−5.06, −3.06)	1.27			
	Lohman Indian RSR	2	99.3	−3.19 (−5.15, −1.24)	1.98			
	*Grower-Finisher Phase (22–42 days)*							
ADFI, g/d	Cobb 500	8	99.7	139.58 (43.64, 235.50)	19,111.51	3	13.11	0.0044
	Ross 308	12	99.0	10.46 (0.16, 20.76)	328.99			
	Ross	5	94.8	−1.11 (−1.92, −0.29)	0.80			
ADG, g/d	Cobb 500	3	64.7	24.91 (23.24, 26.57)	1.39	2	975.23	<0.0001
	Ross	5	95.1	−0.52 (−120, 0.16)	0.54			
BWG, g	Cobb 500	6	99.3	19.74 (13.07, 26.41)	68.90	3	25.56	<0.0001
	Ross 308	11	99.4	4.16 (0.56, 7.76)	36.26			
	Ross	3	99.7	3.28 (−273.99, 842.31)	242,927.9			
FCR, g/g	Cobb 500	6	99.7	−8.39 (−11.19, −5.59)	12.15	3	442.11	<0.0001
	Ross 308	12	98.7	−4.72 (−7.63, −1.80)	25.67			
	Ross	5	17.4	−0.13 (4.51, 5.46)	0.0044			

NC: the number of comparisons between exogenous phytase treatment and deficient P and Ca control group; *I*^2^: proportion of total variation in the common effect size estimate due to heterogeneity; SMD: standardized mean difference; CI: confidence interval; Q: X^2^ statistics; d.f: degree of freedom; τ^2^: heterogeneity variance of the actual effect. ^a^ *p*-value to test for subgroup differences (random effect).

**Table 7 animals-14-02090-t007:** Meta-ANOVA of broiler strain association with measured bone outcomes.

			Random Effects Model					
Parameter	Covariate: Strain	NC	*I*^2^, %	SMD (95% CI)	τ^2^	d.f	Q	*p*-Value ^a^
	*Starter Phase (1–22 days)*							
BBS, Kgf/mm	Cobb 500	10	93.5	4.12 (1.63, 6.60)	15.31	1	0.50	0.4776
	Ross 308	7	91.1	5.52 (2.58, 8.46)	14.56			
Tibia Ash, g/Kg DM	Ross 708	3	89.2	4.11 (2.42, 5.79)	1.92	5	20.68	0.0009
	Cobb 500	25	99.4	11.66 (7.84, 15.48)	93.25			
	Ross × Ross 708	3	99.9	5.02 (−8.25, 18.28)	137.28			
	Ross 708 × YPM	3	0.0	4.88 (2.83, 6.92)	0.00			
	Ross 308	16	95.5	19.00 (9.39, 28.62)	366.62			
Tibia Ca, g/Kg DM	Cobb 500	12	96.5	1.84 (−2.00, 5.68)	45.13	1	5.38	0.0204
	Ross 308	5	97.3	10.50 (4.27, 16.72)	48.66			
Tibia P, g/Kg DM	Cobb 500	13	97.8	2.39 (−0.99, 5.77)	37.91	1	17.67	<0.0001
	Ross 308	5	90.4	12.72 (9.29, 16.15)	13.14			
	*Grower-Finisher Phase (22–42 days)*							
BBS, Kgf/mm	Ross 308	8	96.2	4.88 (2.49, 7.27)	10.91	1	1.73	0.19
	Cobb 500	3	68.7	3.07 (1.83, 4.32)				
Tibia Ash, g/Kg DM	Ross 308	7	93.8	2.38 (−0.13, 4.89)	10.87	3	68.22	<0.0001
	Ross 708	3	65.7	10.87 (8.60, 13.14)	2.58			
	Cobb 500	6	91.3	13.00 (3.53, 22.46)	133.92			
	Ross	5	49.4	1.22 (0.51, 1.94)	0.32			
Tibia Ca, g/Kg DM	Ross 308	5	88.3	9.20 (1.27, 17.13)	77.12	3	10.84	0.0044
	Cobb 500	3	27.5	6.28 (1.87, 10.70)	14.05			
	Ross	5	95.6	0.55 (0.02, 1.09)	0.10			
Tibia P, g/Kg DM	Cobb 500	3	93.6	9.20 (3.70, 14.69)	21.37	2	13.41	0.0012
	Ross	5	89.4	0.32 (−1.33, 1.97)	3.17			
	Ross 308	5	95.5	9.61 (1.59, 17.62)	78.58			

NC: the number of comparisons between exogenous phytase treatment and deficient P and Ca control group; *I*^2^: proportion of total variation in the common effect size estimate due to heterogeneity; SMD: standardized mean difference; CI: confidence interval; Q: X^2^ statistics; d.f: degree of freedom; τ^2^: heterogeneity variance of the actual effect. ^a^ *p*-value to test for subgroup differences (random effects).

**Table 8 animals-14-02090-t008:** Meta-ANOVA of basal diet’s dietary P source association with measured growth outcomes.

			Random Effects Model					
Parameter	Covariate: Dietary P Source ^a^	NC	*I*^2^, %	SMD (95% CI)	τ^2^	d.f	Q	*p*-Value ^b^
	*Starter Phase (1–22 days)*							
ADFI, g/d	Corn–Soybean	69	99.6	15.33 (7.03, 23.62)	1233.91	5	18.52	0.0024
	Wheat–Corn–Soybean	9	99.6	1.86 (−0.24, 0.03)	2249.25			
	CSRRB	2	99.2	6.56 (3.27, 9.86)	5.44			
	Corn–Soybean–DDGS	3	98.9	8.99 (2.54, 15.45)	31.98			
	Wheat–Soybean	11	99.8	10.10 (4.92, 15.28)	76.63			
	WRES	3	95.9	−0.32 (−4.88, 4.24)	15.68			
ADG, g/d	Corn–Soybean	6	99.8	9.31 (1.03, 17.59)	106.92	2	36.83	<0.0001
	Wheat–Corn–Soybean	3	99.4	−0.31 (−3.06, 2.43)	5.86			
	Wheat–Soybean	4	99.4	23.02 (15.74, 30.31)	54.77			
BWG, g	Corn–Soybean	72	99.6	11.35 (7.99, 14.70)	209.70	5	5.89	0.3167
	Wheat-Corn–Soybean	6	99.5	6.97 (2.06, 11.88)	37.48			
	CSRRB	2	90.5	8.61 (6.00, 11.23)	3.23			
	Corn–Soybean–DDGS	3	70.9	8.17 (6.84,9.51)	0.98			
	Wheat–Soybean	9	99.8	11.79 (4.20, 19.37)	134.58			
	WRES	4	92.8	6.40 (3.38,9.41)	8.25			
FCR, g/g	Corn–Soybean	63	99.5	−10.79 (−16.00, 5.58)	442.82	6	488.97	<0.0001
	Wheat–Corn–Soybean	9	99.6	−4.35 (−7.80, −0.91)	27.58			
	CSRRB	2	0.0	−12.04 (−13.12, −10.95)	0			
	Corn–Soybean–DDGS	3	78.9	−3.31 (−4.08, −2.54)	0.36			
	Wheat–Soybean	7	99.5	−4.85 (−7.07, −2.64)	8.91			
	WRES	4	73.8	−8.21 (−10.89, −5.54)	5.41			
	Corn–Soybean	2	0.0	−0.50 (−0.75, −0.24)	0			
	*Grower-Finisher Phase (22–42 days)*							
ADFI, g/d	Corn–Soybean	13	99.8	86.28 (17.50, 155.05)	15,976.48	3	30.93	<0.0001
	Wheat–Corn–Soybean	6	93.5	−1.10 (−1.76, −0.44)	0.62			
	Corn–Soybean–DDGS	3	96.9	39.35 (23.13, 55.57)	196.29			
	WRES	4	95.1	0.62 (−2.75, 3.99)	11.36			
ADG, g/d	Corn–Soyabean	4	99.9	18.18 (4.89, 31.47)	183.31	1	7.59	0.0059
	Wheat–Corn–Soybean	5	95.1	−0.52 (−120, 0.16)	0.54			
BWG, g	Corn–Soybean	11	99.8	11.61 (5.00, 18.22)	124.72	3	19.30	0.0002
	Wheat–Corn–Soybean	4	98.9	−2.38 (−6.20, 1.43)	15.02			
	Corn–Soybean–DDGS	3	98.5	7.91 (2.41, 13.42)	23.19			
	WRES	4	90.1	6.43 (2.59, 10.26)	13.61			
FCR, g/g	Corn–Soybean	11	99.7	−4.73 (−7.89, −1.58)	28.44	3	17.94	0.0005
	Wheat–Corn–Soybean	6	98.7	−2.66 (−7.42, 2.09)	35.21			
	Corn–Soybean	3	81.9	−0.91 (−1.49, −0.34)	0.21			
	WRES	4	93.2	−8.30 (−12.36, −4.25)	14.57			

NC: the number of comparisons between exogenous phytase treatment and deficient P and Ca control group; *I*^2^: proportion of total variation in the common effect size estimate due to heterogeneity; SMD: standardized mean difference; CI: confidence interval; Q: X^2^ statistics; d.f: degree of freedom; τ^2^: heterogeneity variance of the actual effect. ^a^ WRES: wheat–rapeseed expeller–soybean; CSRRB: corn–soybean–rapeseed–rice bran. ^b^ *p*-value to test for subgroup differences (random effects).

**Table 9 animals-14-02090-t009:** Meta-ANOVA of basal diet’s sources of phosphorus association with measured bone outcomes.

			Random Effects Model					
Parameter	Covariate: Dietary P Source ^a^	NC	*I*^2^, %	SMD (95% CI)	τ^2^	d.f	Q	*p*-Value ^b^
	*Starter Phase (1–22 days)*							
BBS, Kgf/mm	Corn–Soybean	13	91.9	3.65 (1.78, 5.51)	11.22	1	4.55	0.0330
	Wheat–Soybean	4	90.5	8.16 (4.45, 11.88)	12.27			
Tibia Ash, g/Kg DM	Corn–Soybean	45	99.4	4.11 (2.42, 5.79)	1.92	2	38.83	<0.0001
	Wheat–Soybean	4	89.9	36.74 (30.47, 43.02)	29.80			
Tibia Ca, g/Kg DM	Corn–Soybean	15	97.4	4.52 (0.27, 8.76)	69.12	1	0.45	0.5041
	Wheat–Corn–Soybean	2	0.0	3.05 (2.39, 3.73)	0			
Tibia P, g/Kg DM	Corn–Soybean	16	98.0	4.65 (0.99, 8.31)	54.70	1	2.04	0.1534
	Wheat–Corn–Soybean	2	92.4	10.14 (3.55, 16.73)	20.91			
	*Grower-Finisher Phase (22–42 days)*							
BBS, Kgf/mm	Corn–Soybean	9	92.7	3.59 (2.27, 4.91)	3.63	1	16.33	<0.0001
	WRES	2	0.0	9.03 (6.75, 11.32)				
Tibia Ash, g/Kg DM	Corn–Soybean	16	97.8	7.53 (3.96, 11.09)	50.60	1	11.54	0.0007
	Wheat–Corn–Soybean	5	49.4	1.22 (0.51, 1.94)	0.32			
Tibia Ca, g/Kg DM	Corn–Soybean	5	95.2	3.73 (−0.08, 7.55)	18.188	2	68.87	<0.0001
	Wheat–Corn–Soybean	5	27.5	0.55 (0.02, 1.09)	0.10			
	WRES	3	13.7	15.44 (11.91, 18.97)	2.02			
Tibia P, g/Kg DM	Corn–Soybean	5	96.0	5.55 (0.36, 10.74)	33.66	2	72.40	<0.0001
	Wheat–Corn–Soybean	5	89.4	0.32 (−1.33, 1.97)	3.17			
	WRES	3	0.0	16.12 (12.86, 19.38)	0			

NC: the number of comparisons between exogenous phytase treatment and deficient P and Ca control group; *I*^2^: proportion of total variation in the common effect size estimate due to heterogeneity; SMD: standardized mean difference; CI: confidence interval; Q: X^2^ statistics; d.f: degree of freedom; τ^2^: heterogeneity variance of the actual effect. ^a^ WRES: wheat–rapeseed expeller–soybean; CSRRB: corn–soybean–rapeseed–rice bran. ^b^ *p*-value to test for subgroup differences (random effects).

**Table 10 animals-14-02090-t010:** Meta-regression of phytase dosage effects on growth, bone strength, and mineralization outcomes.

Dependent Parameter	Intercept	SE (*p*-Value ^a^)	Dosage	QM	d.f.	*p*-Value ^b^	*R*^2^ (%)
	*Starter Phase (1–22 days)*						
ADFI, g/d	13.1238	3.27 (<0.0001)	−0.0003	0.2153	1	0.6426	0.00
ADG, g/d	9.6103	5.82 (0.0985)	0.0016	0.1300	1	0.7185	0.00
BWG, g	10.7360	1.44 (<0.0001)	0.0000	0.0015	1	0.9692	0.00
FCR, g/g	−7.9007	1.95 (<0.0001)	−0.0006	2.4100	1	0.1206	1.32
BBS, Kgf/mm	6.2370	1.86 (0.0008)	−0.0010	0.9577	1	0.3278	0.00
Tibia Ash, g/Kg DM	10.2937	2.86 (0.0003)	0.0015	0.3922	1	0.5311	0.00
Tibia Ca, g/Kg DM	7.9141	3.07 (0.0100)	−0.0029	2.0975	1	0.1475	5.78
Tibia P, g/Kg DM	9.5629	2.69 (0.0004)	−0.0032	3.9557	1	0.0467	15.50
	*Grower-Finisher Phase (22–42 days)*						
ADFI, g/d	38.0642	25.93 (0.1421)	0.0037	0.0332	1	0.8555	0.00
ADG, g/d	9.8424	10.87 (0.3652)	−0.0033	0.0436	1	0.8345	0.00
BWG, g	62.2857	54.22 (0.2507)	−0.0167	0.1632	1	0.6862	0.00
FCR, g/g	−4.5661	1.51 (0.0025)	0.0003	0.0647	1	0.7992	0.00
BBS, Kgf/mm	6.8364	2.03 (0.0008)	−0.0031	1.6752	1	0.1956	3.38
Tibia Ash, g/Kg DM	0.6803	3.24 (0.8339)	0.0063	3.1101	1	0.0778	3.15
Tibia Ca, g/Kg DM	5.0380	4.31 (0.2425)	−0.0001	0.0002	1	0.9886	0.00
Tibia P, g/Kg DM	5.3695	4.81 (0.2646)	0.0005	0.0078	1	0.9297	0.00

QM: coefficient of moderators; QM is considered significant at *p*-value (≤0.05); d.f: degree of freedom; *R*^2^: the amount of heterogeneity accounted for by the covariate (moderator); SE: standard error of intercept. ^a^ *p*-value to the regression model intercept (random effects). ^b^ *p*-value to the regression model moderator (dosage).

**Table 11 animals-14-02090-t011:** Meta-regression of duration of phytase exposure on growth, bone strength, and mineralization outcomes.

Dependent Parameter	Intercept	SE (*p*-Value ^a^)	Duration	QM	d.f.	*p*-Value ^b^	*R*^2^ (%)
	*Starter Phase (1–22 days)*						
ADFI, g/d	−15.3207	15.33 0.3177)	1.5427	3.4411	1	0.0636	2.50
ADG, g/d	−16.1756	35.67 (0.6502)	1.3331	0.5988	1	0.4390	0.00
BWG, g	1.3602	6.32 (0.8296)	0.5302	2.3123	1	0.1284	1.31
FCR	−29.0970	8.31 (0.0005)	1.1359	6.14	1	0.0132	5.28
BBS, Kgf/mm	−57.8350	50.41 (0.2512)	3.0031	1.5383	1	0.2149	1.95
Tibia Ash, g/Kg DM	6.2928	8.19 (0.4423)	0.2953	0.3922	1	0.4965	0.00
Tibia Ca, g/Kg DM	19.4068	5.85 (0.0009)	−0.9962	312.37	1	0.0074	29.05
Tibia P, g/Kg DM	24.2718	4.17 (<0.0001)	−1.2321	22.8293	1	<0.0001	58.00
	*Grower-Finisher Phase (22–42 days)*						
ADFI, g/d	−91.3932	144.35 (0.5266)	6.9971	0.8589	1	0.3540	0.00
ADG, g/d	−60.0350	66.49 (0.3665)	3.5078	1.0447	1	0.3067	0.49
BWG, g	511.5840	267.52 (0.0558)	−24.73	3.0712	1	0.0797	9.11
FCR	−5.7871	8.25 (0.4830)	0.0787	0.0331	1	0.8557	0.00
BBS, Kgf/mm	7.7524	3.53 (0.0282)	−0.1255	0.9370	1	0.3331	0.00
Tibia Ash, g/Kg DM	14.4573	5.32 (0.0065)	−0.3716	2.8099	1	0.0937	10.34
Tibia Ca, g/Kg DM	−43.7178	16.60 (0.0084)	2.5215	8.5614	1	0.0034	40.79
Tibia P, g/Kg DM	−56.0797	16.19 (0.0005)	3.2005	14.4598	1	0.0001	56.07

QM: coefficient of moderators; QM is considered significant at *p*-value (≤0.05); d.f: degree of freedom; *R*^2^: the amount of heterogeneity accounted for by the covariate (moderator); SE: standard error of intercept. ^a^ *p*-value to the regression model intercept (random effects). ^b^ *p*-value to the regression model moderator (dosage).

## Data Availability

The datasets used and analyzed during the current study are available from the corresponding author upon reasonable request.

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
