# Peer review of "Unlocking Phytate with Phytase: A Meta-Analytic View of Meat-Type Chicken Muscle Growth and Bone Mineralization Potential"

_animals, 2024, doi:10.3390/ani14142090_

Round 1
Reviewer 1 Report
Comments and Suggestions for Authors
It is a well-written manuscript containing valuable information on the effects of phytase on growth performance and bone mineralization. The manuscript contains sufficient background information of importance of phyase and bone health/mineralization, detailed methods, clear results, and relevant discussion/conclusion.
Bone ash concentration is one of the critical parameters in bone health. Additional meta-analysis data on bone ash concentration would strengthen this manuscript. In addition, the correlations between growth performance and bone parameters and/or among the bone parameters would give more insight on the beneficial effects of phytase on growth performance and bone development/mineralization in broilers
Author Response
Response to Reviewer 1 Comments
|
|||
1. Summary |
|
|
|
Thank you very much for taking the time to review this manuscript. Please find the detailed responses below and the corresponding revisions/corrections highlighted in the re-submitted files.
|
|||
2. Point-by-point response to Comments and Suggestions for Authors |
|
|
|
Comments 1: Bone ash concentration is one of the critical parameters in bone health. Additional meta-analysis data on bone ash concentration would strengthen this manuscript. |
|||
Response 1: We appreciate you for pointing this out. We agree with this comment. In our global study (Table 4, page 12), subgroup analyses (Table 7-9, lines 524-622), and meta-regression (Table 10 and 11, lines 621-659), we specifically evaluated the ash of the tibia bone. Therefore, an additional meta-analysis on bone ash concentration was not done as suggested. The reason for our decision is that both bone ash and tibia ash are valuable in assessing bone health and mineral content. Whereas bone ash is used to evaluate the overall mineral content of the skeletal system, tibia ash is often preferred in studies where leg strength and integrity are of primary concern in poultry studies. This is because tibia ash provides insights into the localized effects of dietary and environmental factors on the leg bones, which are critical for mobility and structural support.
|
|||
Comments 2: In addition, the correlations between growth performance and bone parameters and/or among the bone parameters would give more insight on the beneficial effects of phytase on growth performance and bone development/mineralization in broilers. |
|||
Response 2: Regarding your suggestion to conduct further a correlational analysis between the growth performance and bone health-dependent variables evaluated in this study, we have carefully considered your recommendation. However, this additional analysis does not align with our primary research objectives or hypothesis set for the study. Our research is specifically focused on 1) summarizing the size of the supplemented exogenous phytase effect on growth performance, bone strength, and mineralization in broilers fed P and Ca deficient basal diets and 2) estimating and exploring the heterogeneity of the effect sizes of outcomes using subgroup and meta-regression analyses. Hence, we hypothesized that reducing P and Ca in a diet supplemented with phytase would not compromise broilers' performance, bone strength, and mineralization at different growth phases. The proposed analysis would extend beyond the scope of our intended investigation.
|
|||
- Response to Comments on the Quality of English Language
Response: The quality of the English Language in our manuscript has been checked by the first author, who is fluent and conversant in English writing.
- Additional clarifications
We would like to clarify further changes in some parts of the manuscript. Some changes have been made in the abstract, introduction, materials and methods, results, and discussion sections of the manuscript to reflect the similarity rate recommendation by the Editor. Secondly, we would like to state that a simple summary on page one, lines 17-31, has been included in the manuscript based on the requirement of the Journal.
Reviewer 2 Report
Comments and Suggestions for Authors
article regarding phytase effect using meta analytic approach is of interest.
Therefore, authors achieved in obvious conclusions on Phytase effect on performance and bone mineralization. None of the data are associated with specific dose. Finaly, phytate and phytase relationship are not given anywhere in the publication.
Author Response
1. Summary |
|
|
|
Thank you very much for taking the time to review this manuscript. Please find the detailed responses below and the corresponding revisions/corrections highlighted in the re-submitted files.
|
|||
2. Questions for General Evaluation |
Reviewer’s Evaluation |
Response and Revisions |
|
Does the introduction provide sufficient background and include all relevant references?
Are the results clearly presented?
Are the conclusions supported by the results? |
Can be improved
Can be improved
Must be improved |
We appreciate you for pointing this out. However, we hold a different view to this suggestion. The introduction sufficiently placed the study in a broad context and highlighted its importance. It further defined the purpose of the study and its significance. The current state of phytase supplementation in broiler chicken was carefully reviewed, and key publications were cited. Thank you for pointing this out. Having conducted sixteen different meta-analyses, presenting the results as a forest plot would have made the manuscript lengthy and the comparison of the physiological phase not clear. Due to this reason, the findings of our forest plots were summarized using tables, and the respective forest plots were presented as supplementary materials. Thank you for pointing this out. The conclusions have been modified to support the results of our review. These modifications have been addressed accordingly in the conclusions section on page 25, lines 860-863. |
|
3. Point-by-point response to Comments and Suggestions for Authors |
|||
Comments 1: Article regarding phytase effect using meta analytic approach is of interest. Therefore, authors achieved in obvious conclusions on Phytase effect on performance and bone mineralization. |
|||
Response 1: Thank you for pointing this out. Although some of the findings of our review confirmed the obvious in terms of phytase supplementation effects on broiler growth performance and bone mineralization, interestingly, we found in our analysis and evaluation that the effects of phytase are significantly correlated with age, broiler strain (Cobb 500 and Ross 308), dietary P source (corn-soybean, wheat-soybean, and WRES-based diets) and extended duration of phytase supplementation rather than phytase dosage in our subgroup and meta-regression analysis. This refutes the assertion that the adverse effects of low P and Ca diets on growth performance and bone mineralization of broilers are possibly ameliorated by superdosing phytase in the diet. Increasing phytase dose beyond commercial recommendations did not influence outcomes of interest significantly.
|
|||
Comments 2: None of the data are associated with specific dose. |
|||
Response 2: We appreciate you for drawing our attention to this. After careful consideration, we disagree with the comment because Table 10 (page 19, lines 621-622) clearly presented data associated with phytase dose (a covariate) in our meta-regression analysis across the growth phase and dependent variables summarized and evaluated in our study. Whereas phytase significantly correlated with Tibia P in the starter phase (depicted in Figure 3 using a bubble plot), increasing the dosage of phytase non-significantly correlated with all other variables evaluated.
Comments 3: Finaly, phytate and phytase relationship are not given anywhere in the publication. Response 3: Agree. Accordingly, we modified the manuscript to emphasize the relationship between phytate and phytase activity. The revised manuscript has this modification on page number 22, paragraph 1, lines 672-674.
3. Response to Comments on the Quality of English Language Response: The quality of the English Language in our manuscript has been checked by the first author, who is fluent and conversant in English writing.
4. Additional clarifications We would like to clarify further changes in some parts of the manuscript. Some changes have been made in the abstract, introduction, materials and methods, results, and discussion sections of the manuscript to reflect the similarity rate recommendation by the Editor. Secondly, we would like to state that a simple summary on page one, lines 17-31, has been included in the manuscript based on the requirement of the Journal.
|
|||
Reviewer 3 Report
Comments and Suggestions for Authors
The current article on the Phytase and Phytate is of researchers interest. Paper contains fruitful information on the subject and will have significant contribution to the broiler feed industry and researchers.
The article is well written over all. The abstract has enough information. Introduction is well explained and all the rest parts are written well.
In abstract, and in conclusion, it is written that phytase reduces FCR. It is confusing. It is better to use word phytase improves the FCR. Please read and correct.
It may need little improvement in English Language.
Comments on the Quality of English LanguageThe same as above.
Author Response
1. Summary |
|
|
|
Thank you very much for taking the time to review this manuscript. Please find the detailed responses below and the corresponding revisions/corrections highlighted in the re-submitted files.
|
|||
2. Point-by-point response to Comments and Suggestions for Authors |
|
|
|
Comments 1: In abstract, and in conclusion, it is written that phytase reduces FCR. It is confusing. It is better to use word phytase improves the FCR. Please read and correct. |
|||
Response 1: Thank you for pointing this out. We agree with this comment. Therefore, we have modified the manuscript in the abstract and conclusions for clarity. The revision was made on page 1, line 45, for the abstract, and on page 25, line 854, for conclusions of the revised manuscript. |
|||
2. Response to Comments on the Quality of English Language Comment 2: It may need little improvement in English Language. Response 2: The quality of the English Language in our manuscript has been checked by the first author, who is fluent and conversant in English writing.
|
|||
- Additional clarifications
We would like to clarify further changes in some parts of the manuscript. Some changes have been made in the abstract, introduction, materials and methods, results, and discussion sections of the manuscript to reflect the similarity rate recommendation by the Editor. Secondly, we would like to state that a simple summary on page one, lines 17-31, has been included in the manuscript based on the requirement of the Journal.
Reviewer 4 Report
Comments and Suggestions for Authors
Phytase is used in chicken diet successfully. However, phytase is not only improve the utization of p, but also, improve the digestibility of CP also. This manuscript tried to review the effect of phytase in the diet of broilers.However, there are some mistakes in the Materials and Methods
1. 124-128 “literature search for articles published between 2000 and February 2024 was conducted using Web of Science (accessed on 21 February 2024), Scopus (accessed on 22 February 2024), ScienceDirect (accessed on 21 February 2024), PubMed (accessed on 22 February 2024), Poultry Science (accessed on 21 February 2024), and Google Scholar (accessed on 20 February 2024) online databases“ . However, as a literature method, Web of Science , Scopus, ScienceDirect , PubMed is some database, however, poultry science is one of the journal. If poultry science should be listed, British poultry science, Applied poultry research should be listed also.
2. 128-130 keywords “phytase supplementation”, “phosphorus”, “broiler chicken”, “growth”, “bone mineralization”, and “blood characteristics” were used. The title “Meat-type Chicken Muscle Growth and Bone Mineralization Potential”. The key words is not consistent with the title.
3. In Table 3 the number of ADG, g/day is noly 6, ADG is a very important parameter in broiler chickens. In Table 4, the number of Tibia Ca, g/Kg DM is only 4. I don’t think it is representative.
4. how the dosage of phytase and the effects of performance, bone and muscle growth is not answered.
Based on not enough data, the authors had better focus on broiler performance and Bone mineralization.
Author Response
1. Summary |
|
|
|
Thank you very much for taking the time to review this manuscript. Please find the detailed responses below and the corresponding revisions/corrections highlighted in the re-submitted files.
|
|||
2. Questions for General Evaluation |
Reviewer’s Evaluation |
Response and Revisions |
|
Does the introduction provide sufficient background and include all relevant references? Is the research design appropriate?
Is the research design appropriate
Are the methods adequately described?
Are the results clearly presented?
Are the conclusions supported by the results? |
Can be improved
Must be improved
Must be improved
Can be improved
Can be improved |
We appreciate you for pointing this out. However, we hold a different view to this suggestion. The introduction sufficiently placed the study in a broad context and highlighted its importance. It further defined the purpose of the study and its significance. The current state of phytase supplementation in broiler chicken was carefully reviewed, and key publications were cited. Thank you for your valuable feedback on our manuscript. We appreciate your concern regarding the appropriateness of our research design. After careful consideration, the design, as currently outlined, is well-suited to meet the objectives of our study. We appreciate your suggestion. The materials and methods section is comprehensive and meets all the elements that ensure thoroughness and repetition of the study. Thank you for pointing this out. Having conducted sixteen different meta-analyses, presenting the results as a forest plot would have made the manuscript lengthy and the comparison of the physiological phase not clear. Due to this reason, the findings of our forest plots were summarized using tables, and the respective forest plots were presented as supplementary materials. Thank you for pointing this out. The conclusions have been modified to support the results of our review. These modifications have been addressed accordingly in the conclusions section on page 25, lines 860-863. |
|
3. Point-by-point response to Comments and Suggestions for Authors |
|||
Comments 1: 124-128 “literature search for articles published between 2000 and February 2024 was conducted using Web of Science (accessed on 21 February 2024), Scopus (accessed on 22 February 2024), ScienceDirect (accessed on 21 February 2024), PubMed (accessed on 22 February 2024), Poultry Science (accessed on 21 February 2024), and Google Scholar (accessed on 20 February 2024) online databases“. However, as a literature method, Web of Science, Scopus, ScienceDirect, PubMed is some database, however, poultry science is one of the journal. If poultry science should be listed, British poultry science, Applied poultry research should be listed also. |
|||
Response 1: Thank you for pointing this out. We agree with the comment that Poultry Science is a journal and not a database. However, the Poultry Science Journal was used in conjunction with the listed databases in our analysis rather than those suggested because, in poultry research, the Poultry Science Journal is highly influential and a leading journal in this scope of research, which frequently publishes relevant articles. Therefore, we have addressed this comment by modifying the manuscript on page 3, paragraph 2, lines 140-142.
|
|||
Comments 2: 128-130 keywords “phytase supplementation”, “phosphorus”, “broiler chicken”, “growth”, “bone mineralization”, and “blood characteristics” were used. The title “Meat-type Chicken Muscle Growth and Bone Mineralization Potential”. The keywords are not consistent with the title. |
|||
Response 2: Thank you for your valuable comment. We have carefully considered your comment. We want to draw your attention to the fact that the working topic at the onset of our study sought to summarize and evaluate dependent parameters, including growth performance, bone mineralization, and blood characteristics, and so we used the reported keywords on page 3, lines 142-144 in our search. However, after removing duplicates and carefully screening the remaining articles based on the inclusion and exclusion criteria, the blood characteristics data to be extracted from the included studies were insufficient. This led to its removal from our final analysis. Secondly, these (“phytase supplementation”, “phosphorus”, “broiler chicken”, “growth”, “bone mineralization”, and “blood characteristics”) were the keywords used in our search and, therefore, need to be reported as such. In addition, the search keywords are consistent with our final title because 1) broilers are meat-type chicken, 2) muscle growth is a measure of growth performance, 3) bone mineralization is consistent with bone mineralization, 4) phosphorus and phytase supplementation are consistent with unlocking phytate with phytase.
Comments 3: In Table 3 the number of ADG, g/day is noly 6, ADG is a very important parameter in broiler chickens. In Table 4, the number of Tibia Ca, g/Kg DM is only 4. I don’t think it is representative. Response 3: We appreciate your comment. We want to draw your attention to the fact that (n), as indicated in Tables 3 and 4 on pages 10 and 11 accordingly, is not the sample size of the parameter evaluated but instead denotes the number of included studies (papers) in our meta-analysis that reported that parameter of interest. Therefore, n, as explained in the footnote of the Tables, need not be representative of any given population since it was not drawn from a population. In addition, as indicated on page 6, lines 262-264, each treatment comparison with the control was considered a separate trial for studies that supplemented more than one phytase dosage. Despite the number of studies (n) that reported ADG being low, the number of comparisons (NC), which the meta-analysis mainly hinges on, is adequate to provide the overall effects.
Comments 4: How the dosage of phytase and the effects of performance, bone and muscle growth is not answered. Based on not enough data, the authors had better focus on broiler performance and Bone mineralization. Response 4: We appreciate your valuable comment. However, we would like to point out that the point raised in the comment provided has been addressed adequately in our manuscript. Lines 601-629 describe the dosage of phytase effects on growth performance and bone mineralization (Table 10, on page 19). Subsequently, on page 22, lines 759-768 adequately explain the dosage effects. Lastly, our current study did not extend beyond the scope of our intended investigation other than summarizing and evaluating the size of the supplemented phytase effect on growth performance, bone strength, and mineralization in broilers fed P and Ca deficient basal diets and 2) estimating and exploring the heterogeneity of the effect sizes of outcomes using sub-group and meta-regression analyses. |
|||
|
|||
- Response to Comments on the Quality of English Language
Response: The quality of the English Language in our manuscript has been checked by the first author, who is fluent and conversant in English writing.
- Additional clarifications
We would like to clarify further changes in some sections of the manuscript. Some changes have been made in the abstract, introduction, materials and methods, results, and discussion sections of the manuscript to reflect the similarity rate recommendation by the Editor. Secondly, we would like to state that a simple summary on page one, lines 17-31, has been included in the manuscript based on the requirement of the Journal.